# Glypicans define unique roles for the Hedgehog co-receptors boi and ihog in cytoneme-mediated gradient formation

Eléanor Simon[1†], Carlos Jiménez-Jiménez[1†], Irene Seijo-Barandiarán[1], Gustavo Aguilar[1,2], David Sánchez-Hernández[1], Adrián Aguirre-Tamaral[1], Laura González-Méndez[1], Pedro Ripoll[1‡], Isabel Guerrero[1]*

[1]Tissue and Organ Homeostasis, Centro de Biología Molecular "Severo Ochoa" (CSIC-UAM), Nicolás Cabrera 1, Universidad Autónoma de Madrid, Cantoblanco, Spain; [2]Growth and Development, University of Basel, Biozentrum, Switzerland

**Abstract** The conserved family of Hedgehog (Hh) signaling proteins plays a key role in cell–cell communication in development, tissue repair, and cancer progression, inducing distinct concentration-dependent responses in target cells located at short and long distances. One simple mechanism for long distance dispersal of the lipid modified Hh is the direct contact between cell membranes through filopodia-like structures known as cytonemes. Here we have analyzed in *Drosophila* the interaction between the glypicans Dally and Dally-like protein, necessary for Hh signaling, and the adhesion molecules and Hh coreceptors Ihog and Boi. We describe that glypicans are required to maintain the levels of Ihog, but not of Boi. We also show that the overexpression of Ihog, but not of Boi, regulates cytoneme dynamics through their interaction with glypicans, the Ihog fibronectin III domains being essential for this interaction. Our data suggest that the regulation of glypicans over Hh signaling is specifically given by their interaction with Ihog in cytonemes. Contrary to previous data, we also show that there is no redundancy of Ihog and Boi functions in Hh gradient formation, being Ihog, but not of Boi, essential for the long-range gradient.

*For correspondence:
iguerrero@cbm.csic.es

[†]These authors contributed equally to this work
[‡]Retired CSIC Research Professor

**Competing interests:** The authors declare that no competing interests exist.

## Introduction

The Hedgehog (Hh) signaling pathway has a conserved central role in cell–cell communication during tissue patterning, stem cell maintenance, and cancer progression (*Briscoe and Thérond, 2013*). During development, Hh ligand is usually released from a localized source inducing at distance concentration-dependent cellular differentiation and/or proliferation responses. Hh is synthetized as a precursor molecule that undergoes two post-translational lipid modifications: cholesterol (*Porter et al., 1996*) and palmitic acid (*Pepinsky et al., 1998*), which associate Hh to cell membranes restricting its free spreading through the extracellular milieu (*Peters et al., 2004*). Several mechanisms have been proposed to explain Hh dispersion (*Lewis et al., 2001*); among these, the transport by filopodia-like structures (known as cytonemes) is the most suitable model, since they facilitate direct contact between cell membranes for delivery and reception (reviewed in *González-Méndez et al., 2019*).

Cytonemes have been described as dynamic actin-based protrusive structures that deliver or uptake many signaling proteins in different biological contexts (*Ramírez-Weber and Kornberg, 1999*; reviewed in *González-Méndez et al., 2019*). In *Drosophila*, Hh has been shown to be localized along cytonemes in the germline stem cells niche (*Rojas-Ríos et al., 2012*), in abdominal histoblast nests, and in wing imaginal discs (*Callejo et al., 2011*; *Bilioni et al., 2013*; *Gradilla et al., 2014*; *Chen et al., 2017*). Spatial and temporal correlations between Hh gradient establishment and

cytoneme formation have also been described (*Bischoff et al., 2013*; *González-Méndez et al., 2017*). As the *Drosophila* Hh (*Gradilla et al., 2014*), the vertebrate homolog Sonic-hh (Shh) has also been visualized in vivo in vesicle-like structures moving along filopodia protruding from Shh-producing cells during chick limb bud development (*Sanders et al., 2013*), in mouse tissue culture cells (*Hall et al., 2021*), and during regeneration of axolotls (*Zhang et al., 2021*). Although the importance of cytonemes in the coordination of growth and patterning is well documented (*Ali-Murthy and Kornberg, 2016*; *González-Méndez et al., 2019*; *Zhang and Scholpp, 2019*), much less is known about the intrinsic and extrinsic cues that regulate their establishment and dynamics.

In larval wing imaginal discs and pupal abdominal histoblast nests, Hh is produced and secreted by the Posterior (P) compartment cells and transported towards the receiving Anterior (A) compartment cells, resulting in a graded Hh distribution in this compartment. Cytonemes emanating at the basal surface of the polarized epithelia guide Hh delivery directly from P to A at cytoneme contact sites where Hh reception takes place (*Bischoff et al., 2013*; *Chen et al., 2017*; *González-Méndez et al., 2017*; *González-Méndez et al., 2020*). To activate its targets in a concentration-dependent manner, Hh binds to its receptor complex formed by the canonical receptor Patched (Ptc), the co-receptors Interference hedgehog (Ihog) and Brother of Ihog (Boi) (*Yao et al., 2006b*), and the membrane-anchored glypicans Dally-like protein (Dlp) and Dally (*Desbordes and Sanson, 2003*; *Lum et al., 2003b*; *Han et al., 2004*; *Williams et al., 2010*). All these proteins are associated with Hh presenting (*Bilioni et al., 2013*; *Bischoff et al., 2013*) and receiving cytonemes (*Chen et al., 2017*; *González-Méndez et al., 2017*). In addition, Boc and Cdo, homologs of Ihog and Boi in vertebrates, have been reported to be associated to Hh transport and reception through cytonemes (*Sanders et al., 2013*; *Hall et al., 2021*).

The glypicans Dlp and Dally have been described as necessary for cytoneme attachment in the wing imaginal discs (*Huang and Kornberg, 2016*; *González-Méndez et al., 2017*) and for cytoneme formation in the air sac primordium (*Huang and Kornberg, 2016*). Dally and Dlp contain heparan sulfate (HS) and glycosaminoglycan (GAG) chains attached to a core of glypican proteins, which in turn are bound to the membrane via a GPI anchor; they have a generalized, but not uniform, expression in the wing imaginal disc (*Nakato et al., 1995*; *Baeg et al., 2001*). Glypicans regulate most morphogenetic gradients, i.e. Hh (*Han et al., 2004*, reviewed in *Filmus and Capurro, 2014*), Wingless (Wg) (*Franch-Marro et al., 2005*; *Gallet et al., 2008*; *Yan et al., 2009*), Decapentaplegic (Dpp) (*Jackson et al., 1997*; *Belenkaya et al., 2004*; *Norman et al., 2016*), and FGF (*Yan and Lin, 2007*); however, how glypicans achieve their specificity for each signal is not known. In Hh signaling, Dlp is needed for Hh reception (*Desbordes and Sanson, 2003*; *Lum et al., 2003a*; *Han et al., 2004*; *Williams et al., 2010*) and delivery from the Hh-producing cells (*Callejo et al., 2011*), while Dally is needed for maintenance of Hh levels (*Han et al., 2004*; *Bilioni et al., 2013*).

To date, the co-receptors Ihog and Boi are thought to have redundant functions because Hh signaling is only blocked when both are simultaneously absent in the Hh-receiving cells (*Zheng et al., 2010*). Both adhesion proteins share sequence similarities: they are type one transmembrane proteins with four Ig and two Fibronectine type III (FNIII) extracellular domains, and an undefined intracellular domain (*Yao et al., 2006a*). Nevertheless, Ihog and Boi are separately needed for the maintenance of normal Hh levels in Hh-producing cells (*Yan et al., 2010*). Both proteins are expressed ubiquitously in the wing disc, although their protein levels are reduced at the A/P boundary in response to Hh signaling. More specifically, Ihog is more abundant in basal plasma membranes of the wing disc epithelium (*Zheng et al., 2010*; *Bilioni et al., 2013*; *Hsia et al., 2017*) and it has the ability to stabilize cytonemes when overexpressed, clearly influencing cytoneme dynamics (*Bischoff et al., 2013*; *González-Méndez et al., 2017*).

In this work, we further analyze the interactions of glypicans with Ihog and Boi and the role of these interactions on cytoneme dynamics and Hh signaling. We describe that whereas Ihog and Boi functions are not needed for the maintenance of glypicans levels, glypicans are required to maintain Ihog, but not Boi, protein levels. In addition, we observe that ectopic Ihog, but not Boi, stabilizes cytonemes. We further dissect the functional domains of Ihog responsible for the interaction with Dally and Dlp, Hh and Ptc, as well as for cytoneme stability and Hh gradient formation. We describe that the Hh-binding Fn1 domain of Ihog (*McLellan et al., 2006*) is crucial for glypican interaction and cytoneme stabilization, though the amino acids involved are different from those responsible for the Ihog/Hh interaction (*Wu et al., 2019*). Similarly, the previously described Ptc-interacting Fn2 domain (*Zheng et al., 2010*) is also key for Ihog interaction with glypicans as well as for the

stabilization of cytonemes. We conclude that the FNIII domains of Ihog interact with glycans to regulate cytoneme behavior and that these interactions provide glypican specificity for Hh signaling as well as a key function for Hh gradient formation. We further propose that the presence of Ihog, but not of Boi, in basally located cytonemes is essential for Hh signaling gradient formation.

## Results

### Glypicans interact with Ihog to stabilize it at the plasma membranes

Previous work described interactions between glypicans and Ihog that were revealed by the ectopic Ihog recruitment of glypicans in wing disc cells (*Bilioni et al., 2013*). Here, we have further investigated the interactions of Ihog and Boi with glypicans. We did not detect changes to the levels of Dlp and Dally in $boi^{-/-}$ $ihog^{-/-}$ double mutant clones (*Figure 1A,B*). Conversely, Ihog was detectable at abnormally low levels in double mutant clones for *tout velu (ttv)* and *brother of tout velu (botv)* (*Figure 1C*); these genes code for enzymes that synthesize the HS-GAG chains of the glypican core proteins Dally and Dlp (*Takei et al., 2004*; *Han et al., 2004*). While in double $dlp^{-/-}$ $dally^{-/-}$ mutant clones the downregulation is evident (*Figure 1F*), this downregulation is not significant within $dlp^{-/}$ or $dally^{-/-}$ single mutant clones (*Figure 1D,E*). Thus, this effect on Ihog levels only appears in the absence of both Dally and Dlp proteins. This regulation seems to be dose-dependent since in sister clones (homozygous $dlp^{+/+}$ $dally^{+/+}$ or $ttv^{+/+}$ $botv^{+/+}$ wild-type cells contiguous to homozygous mutant clones $dlp^{-/-}$ $dally^{-/-}$ or $ttv^{-/-}$ $botv^{-/-}$) there are higher levels of Ihog than in the heterozygous $dlp^{-/+}$ $dally^{-/+}$ or $ttv^{-/+}$ $botv^{-/+}$ background cells (*Figure 1C,c,F,f*, red arrowheads). Since $ttv^{-/-}$ $botv^{-/-}$ and $dally^{-/-}$ $dlp^{-/-}$ double mutant clones show identical effects on Ihog levels, the interaction between glypicans and Ihog might take place through the HS-GAG chains of the glypicans.

We then analyzed the interaction of glypicans with Boi. Interestingly, in the case of Boi there is no evidence for a similar modulation. In contrast with the strong decrease of Ihog, there are no changes of Boi protein levels *in* $dlp^{-/-}$ $dally^{-/-}$ double mutant clones (*Figure 1G*). In summary, these data reveal a novel functional role of Dally and Dlp to maintain Ihog, but not Boi, protein levels.

We further explored the interaction of Ihog and Boi with glypicans. The apical–basal cellular distribution of Dally and Dlp in the disc epithelium is different: Dlp is more basally distributed than Dally, as it has been previously described (*Gallet et al., 2008*). Since the ectopic Ihog recruits glypicans in wing disc cells, we analyzed if ectopic Boi also does. Overexpressing in the dorsal compartment of the wing disc of Ihog or Boi (*Figure 1—figure supplement 1*), using the ventral compartment as internal control, we observed that although ectopic Boi also accumulates glypicans, the apical/basal distribution of this increase in the disc epithelium was different than that of Ihog. While Boi accumulates Dally and Dlp mainly at the apical side of the disc (*Figure 1—figure supplement 1A*), Ihog accumulates them basally (*Figure 1—figure supplement 1B*). This result, together with the differential effect of Dally and Dlp to maintain Ihog, but not Boi, protein levels at the plasma membrane, suggests that the ways in which Ihog and Boi interact with glypicans differ both in function and in mechanism.

### Ihog functional domains implicated in the glypican–Ihog interaction

To investigate the role of the various Ihog protein domains in Ihog–Dally and Ihog–Dlp interactions, we generated transgenic lines to ectopically express Ihog proteins carrying deletions for the different extracellular domains, each fused to the red fluorescent protein (RFP) in the C-terminal. The following constructs were generated all carrying the transmembrane domain: Ihog lacking the four Ig domains (*UAS.ihogΔIg-RFP*), the two FNIII domains *(UAS.ihogΔFn-RFP)*, the Fn1 domain (*UAS.ihogΔFn1-RFP*), the Fn2 domain (*UAS.ihogΔFn2-RFP);* Ihog containing only the intracellular C-terminus (*UAS.ihogCT-RFP*), or the extracellular domain alone (*UAS.ihogΔCT-RFP)* were also generated. The molecular weights corresponding to the generated Ihog mutant forms were estimated by Western blot (*Figure 2—figure supplement 1*).

Previously, it has been described that ectopic Ihog expression recruits glypicans (*Bilioni et al., 2013*; *Figure 2A,A'*); therefore, the analysis to identify the protein domain(s) responsible for the Ihog–glypican interaction was done through the study of the recruitment of Dally and Dlp after overexpressing the Ihog mutant variants in the dorsal compartment of the wing disc, using the ventral

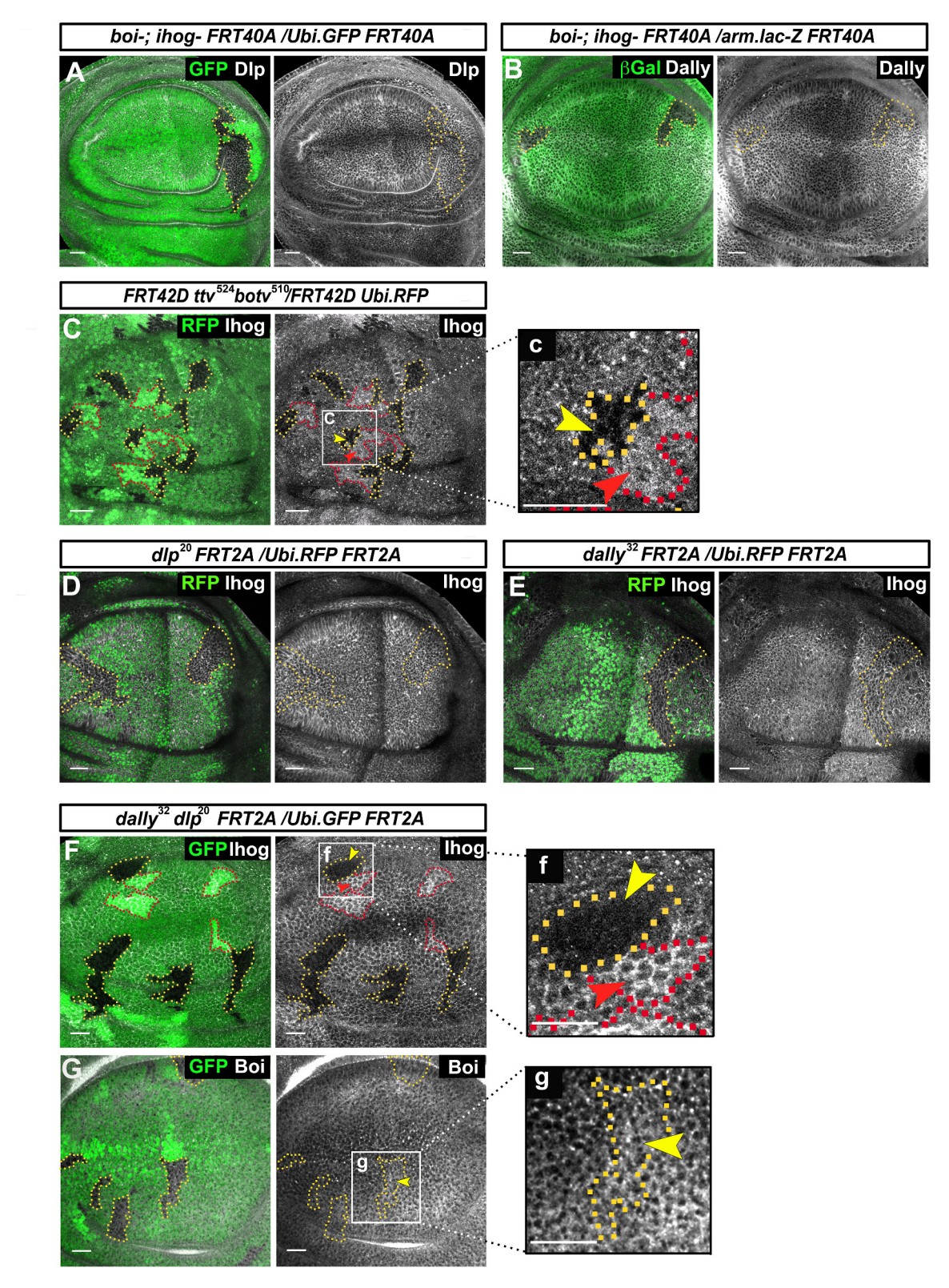

**Figure 1.** Glypicans regulate Ihog but not Boi presence at the plasma membrane. (**A, B**) *boi⁻; ihog^Z23* double mutant clones (labeled by the lack of GFP) do not affect the expression of the glypicans Dlp (**A**) and Dally (**B**) (gray channels). (**C**) Ihog levels (*Bac Ihog:GFP*, gray channel) decrease in *ttv^524 botv^510* double mutant clones (labeled by the lack of GFP, yellow arrowhead); (**c**) enlargement of one of the clones. (**D, E**) *dlp^20* (**D**) and *dally^32* (**E**) single mutant clones (labeled by the lack of RFP in green) do not affect Ihog levels (α-Ihog antibody, gray channel). (**F**) Ihog levels (labeled with α-Ihog antibody, gray

*Figure 1 continued on next page*

*Figure 1 continued*

channel) decrease in *dally*$^{32}$ *dlp*$^{20}$ double mutant clones (labeled by the lack of GFP, yellow arrowhead); (f) enlargement of one clone. (**G**) Boi levels (labeled with anti Boi antibody, gray channel) are not affected in *dally*$^{32}$ *dlp*$^{20}$ double mutant clones (labeled by the lack of GFP, yellow arrowhead); (**g**) enlargement of one clone. Note that the homozygous wild-type sister clones (more intense green) positively modulate Ihog levels (red arrowheads in **C** and **F**). The discs shown in panels are representative of four to eight discs containing clones in at least three experiments. Scale bar: 20 μm.

The online version of this article includes the following figure supplement(s) for figure 1:

**Figure supplement 1.** Different roles of Boi and Ihog in glypican recruitment.

---

compartment as internal control (*Figure 2B–D'*, *Figure 2—figure supplement 2*). The results obtained were quantified (*Figure 2E,E'*), and we found no increase in the amount of either Dally or Dlp after ectopic expression of *UAS.ihogCT-RFP* (*Figure 2—figure supplement 2A,A'*). Ectopic expression of *UAS.ihogΔIg-RFP* increases glypican levels in a way similar to the ectopic expression of the full-length Ihog form (*Figure 2—figure supplement 2B,B'*), while the ectopic *UAS.ihogΔFn-RFP* does not result in Dally or Dlp increase (*Figure 2B,B'*). These results confirm the implication of the extracellular fragment and specifically identify the fibronectin domains as responsible for the glypican interaction. We then determined which of the two FNIII domains was responsible for the Ihog–glypican interaction. Quantification of the increase of glypicans by the expression of *UAS.ihogΔFn1-RFP* shows a very mild increase of Dally and Dlp (*Figure 2C,C' and E,E'*). On the other hand, the expression of *UAS-ihog.ΔFn2-RFP* results in very mild Dally but high Dlp increase (*Figure 2D,D' and E,E'*). These data indicate that while Ihog–Dlp interaction is mediated by the Fn1 domain, Ihog–Dally interaction needs both Fn1 and Fn2 domains.

## The Ihog–Fn1 interacts with Hh and glypicans through different amino acids

Both Ihog and Boi have demonstrated roles in the maintenance of extracellular Hh levels in the Hh-producing cells, since the extracellular levels of Hh decrease in wing discs after the loss of function of Ihog, Boi, or both (*Yan et al., 2010*; *Avanesov and Blair, 2013*; *Bilioni et al., 2013*). According to these data, overexpression of Ihog in the dorsal compartment of the wing disc accumulates Hh mainly at the basolateral plasma membrane of the disc epithelium (*Figure 3A*; *Yan et al., 2010*; *Callejo et al., 2011*; *Bilioni et al., 2013*).

The Ihog–Hh interaction was described to take place through the Ihog-Fn1 domain (*McLellan et al., 2006*; *Yao et al., 2006b*) and, as expected, we found that the expression of either *UAS-ihogΔFn-RFP* or *UAS-ihogΔFn1-RFP* does not result in Hh increase (*Figure 3B* and *Figure 3D*, respectively; *Figure 3G* for quantifications). To further analyze the potential Fn1 domain sequences responsible for Hh and/or glypican accumulation, we made an in silico prediction of the amino-acid residues responsible for the Ihog–Hh interaction based on structural analysis of Ihog (*McLellan et al., 2008*). We then generated an Ihog variant carrying point mutation substitutions in three residues (D558N, N559S, E561Q) in the region previously predicted to interact with Hh via hydrogen bonds (*McLellan et al., 2006*). As anticipated, overexpression of this Ihog form (*UAS. IhogFn1\*\*\*-RFP*) does not accumulate Hh, probably due to loss of the Ihog–Hh interaction when at least one of the three amino acids is mutated (*Figure 3E,G*). We next analyzed the interaction of the same Ihog mutant form, *UAS.IhogFn1\*\*\**, with glypicans and observed that both Dally and Dlp accumulate (*Figure 3H,H'*; *Figure 3I,I'*) at a level similar to that following overexpression of the wild-type Ihog (*Figure 2A,A'*; *Figure 2E,E'*), indicating that the Fn1 amino acids that interact with glypicans are different from those carrying out the Ihog–Hh interaction. To test the sufficiency of Ihog–Fn1 domain for the interaction with Hh, we targeted it to the membrane by fusing it to the extracellular domain of CD8 (*UAS.CD8.Fn1-Cherry*). Expression of this construct in the dorsal compartment resulted in a small increase of Hh levels (*Figure 3F,G*). Altogether these results suggest that although Fn1 is the specific Ihog domain binding Hh and interacting with Dlp and Dally, the optimal binding may also require the cooperation of the Fn2.

## Ihog–Fn2 domain partially influences Hh interaction

The ectopic expression of *UAS-ihog.ΔFn2-RFP* still accumulates Hh (*Figure 3C,G*), although at noticeably lower levels than the expression of the wild-type Ihog (*Figure 3A,G*). This last reduction

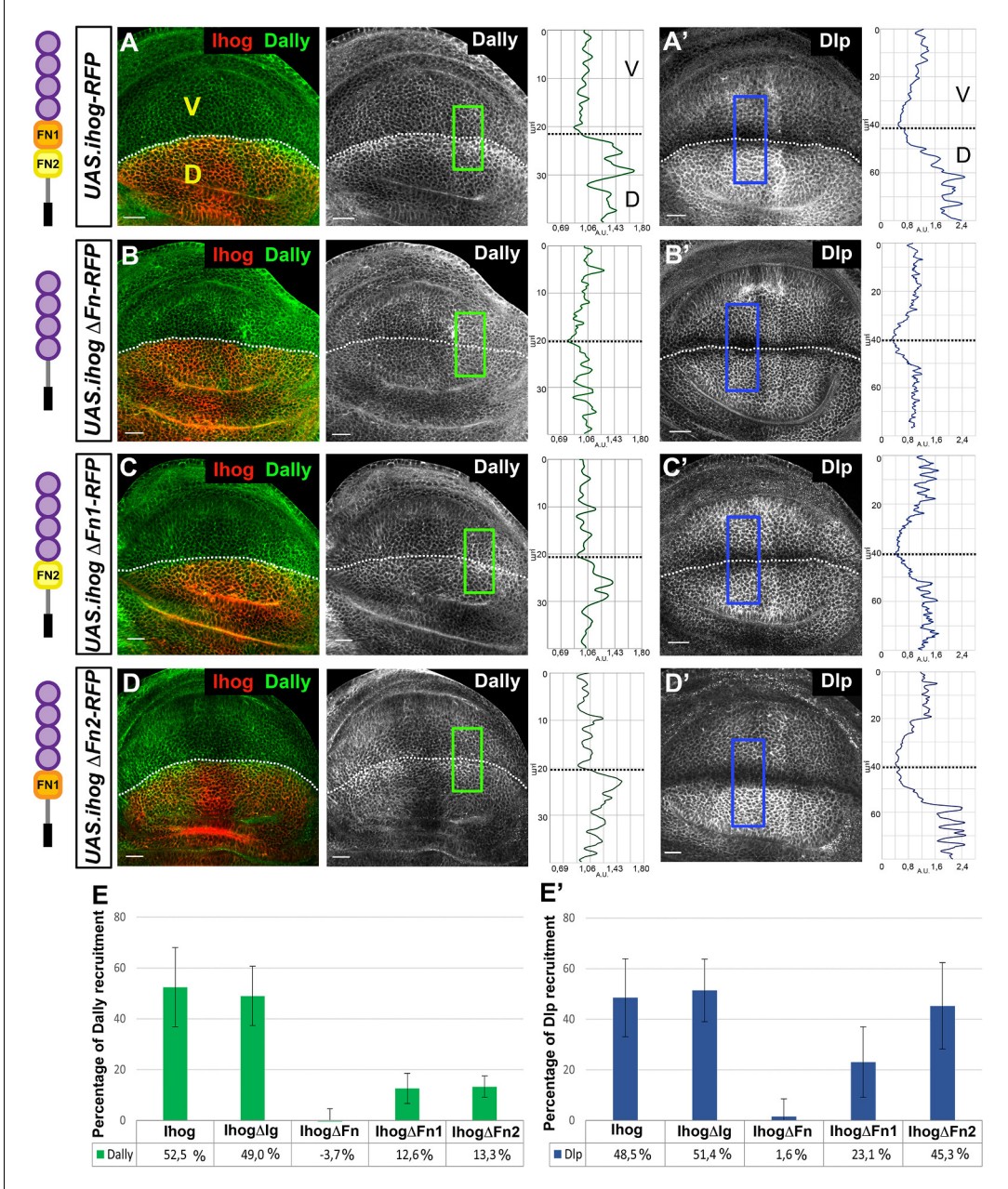

**Figure 2.** Interaction of Ihog mutant forms with glypicans. (A–D) Glypicans accumulation in: *apGal4 tubGal80$^{ts}$/+; UAS.ihog-RFP/+* (A–A'), *apGal4 tubGal80$^{ts}$/+; UAS.ihogΔFn-RFP/ +* (B–B'), *apGal4 tubGal80$^{ts}$/+; UAS.ihogΔFn1-RFP/+* (C–C'), *apGal4tubGal80$^{ts}$/+; UAS.ihogΔFn2-RFP/+* (D–D') wing discs after 30 hr at the restrictive temperature. (E–E') Percentage of Dally (E) and Dlp (E') recruitment in the dorsal compartment relative to the endogenous control (ventral compartment). Each image incorporates at its side a plot profile (taken from the framed area in each image), indicating the relative intensity fluorescence for each glypican. Note that IhogΔFn-RFP does not increase either Dally (B, E) or Dlp (B', E'). However, in ectopic expression of a partial deletions of the Ihog FN-type III domains (IhogΔFn1-RFP and IhogΔFn2-RFP) an accumulation of Dally (C–E) is observed although at lower levels than those of the ectopic Ihog-RFP (A, E). Curiously, IhogΔFn2-RFP recruits Dlp (D', E') at the same levels as Ihog-RFP (A', E') does. Discs are oriented with dorsal part down (D), ventral up (V), and posterior (P) right. Plot profiles were done over a ROI of size of 40 μm X 20 μm for Dally and 80 μm × 20 μm for Dlp. Data of glypicans recruitment is available at *Figure 2—source data 1* for Dally and *Figure 2—source data 2* for Dlp measure; p-values of the statistical analysis are shown in *Tables 1* and *2* (Materials and methods). The discs shown in panels are representative of at least four discs in three independent experiments. Scale bar: 20 μm.

The online version of this article includes the following source data and figure supplement(s) for figure 2:

**Source data 1.** Dally recruitment measures ussing different Ihog mutant constructs.

**Source data 2.** Dlp recruitment measures ussing different Ihog mutant constructs.

*Figure 2 continued on next page*

*Figure 2 continued*

**Figure supplement 1.** Scheme of the Ihog mutant constructs and the molecular weight of their proteins.
**Figure supplement 2.** Role of Ihog ΔIg and Ihog CT domains in glycan retention.

in Hh increase might result from a diminished interaction of Ihog.ΔFn2 with Dally (*Figure 2D,E*), as Dally seems to be needed for Hh stability at the plasma membrane (*Bilioni et al., 2013*). Finally, as expected, Hh recruitment is not observed when expressing only the intracellular part of the protein (*Figure 3—figure supplement 1B*). We therefore conclude that, although the Ihog interaction with Hh is primarily mediated by Fn1, the Fn2 domain may play a minor secondary role as a consequence of its implication in Ihog–Dally interaction.

Previous research has described the role of the Fn2 domain of Ihog as enhancing Hh protein binding in cells co-expressing its receptor Ptc (*Yao et al., 2006b*) through Ptc interaction with two specific Fn2 amino-acid residues (*Zheng et al., 2010*). Since Dlp, and probably Dally, are also part of the Hh reception complex, we next studied if these amino acids could also be implicated in the interaction of Fn2 with Hh and the two glypicans. We generated an Ihog variant carrying point mutations in the two amino-acid residues (K653 and Q655) (*Zheng et al., 2010*) and fused it to RFP (*UAS. ihogFn2\*\*-RFP*). Interestingly, the expression of this Ihog mutant form does not accumulate Hh (*Figure 3—figure supplement 1C*), Dally or Dlp (*Figure 3—figure supplement 1D,D'*). To our surprise, this result is different from that obtained by the expression of *UAS.ihogΔFn2* (*Figure 3C*), lacking the entire Fn2 domain, which still recruits Hh. Moreover, the expression of *UAS.ihogFn2\*\**, with or without RFP (*Zheng et al., 2010*), decreases the amount of endogenous Dally (*Figure 3—figure supplement 1D* and *Figure 3—figure supplement 2A*), acting as dominant negative in the recruitment of Dally and thus indirectly affecting Hh stabilization.

## Ihog Fn1 and Fn2 domains influence Hh gradient formation

To further investigate the effect of *UAS.ihogΔFn2-RFP* and *UAS.ihogFn2\*\*-RFP* variants on Hh reception, we overexpressed the wild-type Ihog and these mutant forms in the Ptc expressing cells of the ventral side of the wing disc, using *LexAop.Gal80; ap.LexA Ptc.Gal4* as driver, thus maintaining the dorsal side as internal wild-type control (*Figure 4*). We then analyzed the response of the high-threshold target Ptc and the low threshold target Cubitus interruptus (Ci). After ectopic Ihog-RFP expression, Hh reception is slightly affected, resulting in flattened gradient responses with an extension of Ptc expression together with a reduction of its highest levels and an extension of Ci expression (*Figure 4A,A'*, see also *Yan et al., 2010*). The flattening effect is more evident when expressing *UAS.ihogΔFn2-RFP* (*Figure 4—figure supplement 1A,A'*). Unexpectedly, no effect is detected after the expression of *UAS-ihogFn2\*\*-RFP*, that carries the two Fn2 point mutations (*Figure 4—figure supplement 1B,B'*) proposed to reduce the interaction with Ptc (*Zheng et al., 2010*).

We then analyzed the role of ΔFn1 in Hh gradient formation. In agreement with the inability of *UAS.ihogΔFn1-RFP* or *UAS.ihogFn1\*\*\*-RFP* to interact with Hh, their ectopic expression in Hh-receiving cells has a different effect, lacking expression of the high-threshold target En (*Figure 4—figure supplement 2A,B*), nearly absent expression of Ptc and expanded expression of the low threshold target Ci (*Figure 4B–C'*). Thus, these Ihog mutants show a dominant negative effect on Hh gradient responses because it should not occur when the endogenous genes *ihog* and *boi* are present.

To test the existence of a dominant negative effect, we used the Ihog Bac-GFP reporter (*Hsia et al., 2017*) and expressed either *UAS.ihogΔFn1-RFP* or *UAS.ihogFn1\*\*\*-RFP*. Under these conditions, the endogenous Ihog levels were clearly diminished mainly at the basal part of the disc epithelium (*Figure 4—figure supplement 3A,B*), supporting the dominant negative effect of these constructs over the endogenous Ihog.

## Excess of ihog, but not of Boi, can modulate cytoneme dynamics

The in vivo imaging of the abdominal histoblast nests allows studying cytoneme dynamics in physiological conditions. In this system, cytoneme behavior is not altered when analyzed using innocuous actin reporters (*Bischoff et al., 2013*; *Chen et al., 2017*; *González-Méndez et al., 2017*) such as the actin-binding domain of moesin fused to GFP (GMA-GFP) (*Bloor and Kiehart, 2001*). Using this

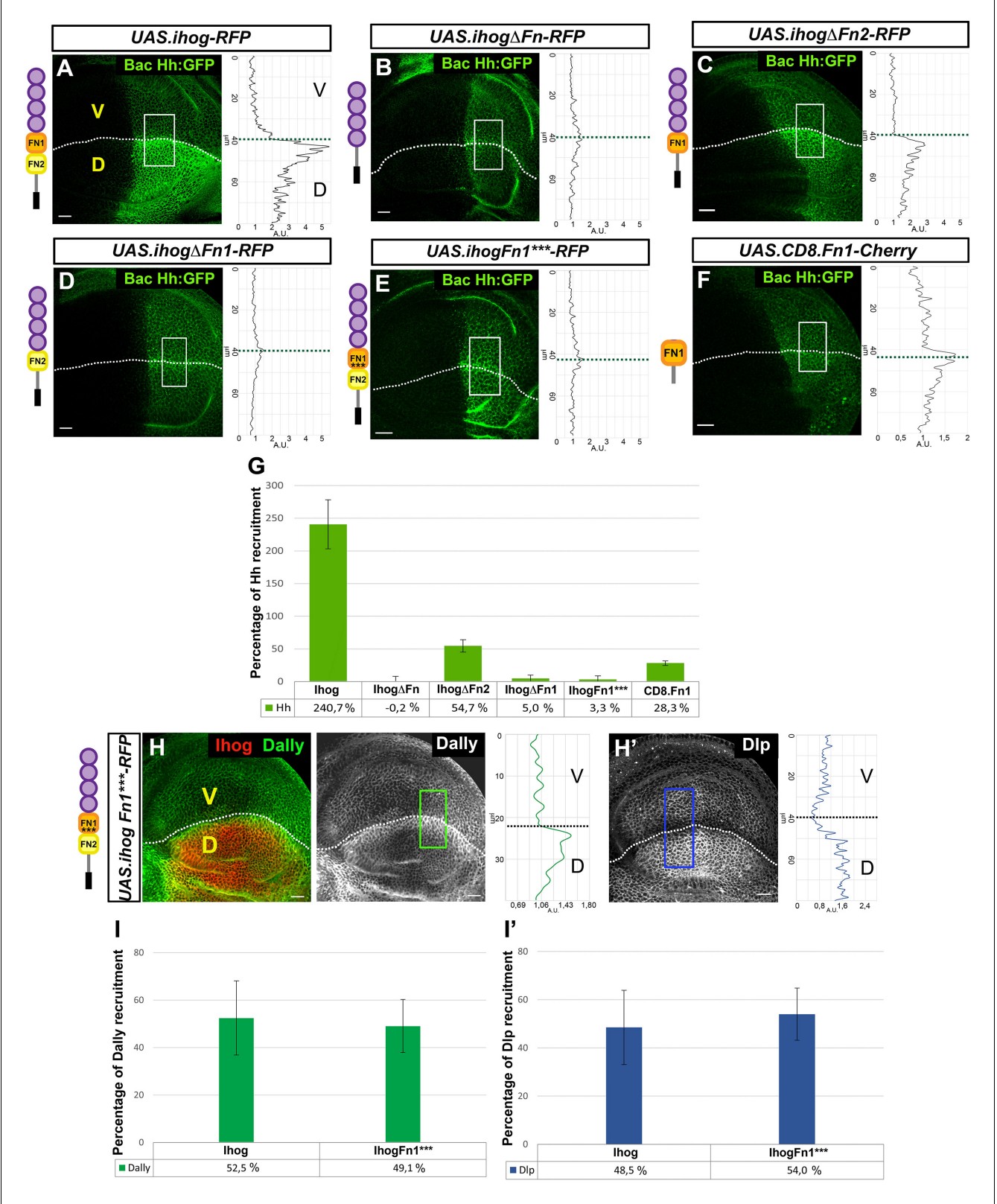

**Figure 3.** Effects of Ihog mutant forms and Ihog Fn point mutations on Hh and glycican interactions. (**A–G**) Hh (BacHh:GFP) expression in wing discs after 30 hr at the restrictive temperature in *apGal4 BacHh:GFP/+; UAS.ihog-RFP/tubGal80$^{ts}$* (**A**), *apGal4 BacHh:GFP/+; UAS.ihogΔFn-RFP/tubGal80$^{ts}$* (**B**), *apGal4 BacHh:GFP/+; UAS.ihogΔFn2-RFP/tubGal80$^{ts}$* (**C**), *apGal4 BacHh:GFP/+; apGal4 BacHh:GFP/+; UAS.ihogΔFn1-RFP/tubGal80$^{ts}$* (**D**), *UAS.*

*Figure 3 continued*

*ihogFn1\*\*\*-RFP/tubGal80$^{ts}$* (E), *apGal4 BacHh:GFP/+; UAS.cd8.Fn1-Cherry/tubGal80$^{ts}$* (F), percentage of Hh recruitment in the dorsal compartment relative to the endogenous control (ventral compartment) (G). (H, I) Increase of glypicans after 30 hr at the restrictive temperature in *apGal4 tubGal80$^{ts}$/+; UAS.ihogFn1\*\*\*-RFP/+* (H, H') and percentage of Dally (I) and Dlp (I') recruitment in the dorsal compartment relative to the endogenous control (ventral compartment). Images incorporate at their side a plot profile (taken from the framed area in each image), indicating the modulation of glypican levels. Plot profiles were done over ROI of size 80 µm × 30 µm for Hh, 40 µm × 20 µm for Dally and 80 µm × 20 µm for Dlp. Note that Ihog Fn1\*\*\* does not interact with Hh (E, G) *while increasing Dally* (H, I) *and Dlp* (H', I') *at same levels that Ihog-RFP does.* Discs are oriented with dorsal part down (D), ventral up (V) and posterior (P) right. Data is available at *Figure 3—source data 1* for Hh recruitment and p-values of the statistical analysis are shown in *Table 3* (Materials and methods). Data is available at *Figure 2—source data 1* for Dally recruitment and *Figure 2—source data 2* for Dlp recruitment; p-values of the statistical analysis are shown in *Tables 1* and *2* (Materials and methods). The discs shown in panels are representative of at least four discs from three independent experiments. Scale bar: 20 µm.

The online version of this article includes the following source data and figure supplement(s) for figure 3:

**Source data 1.** Hh recruitment measures ussing different Ihog mutant constructs.
**Figure supplement 1.** Effects of Ihog mutant forms on Hh and glypicans interactions.
**Figure supplement 2.** Effects of the ectopic expression of IhogFn2\*\*.

system, it has been previously observed that ectopic expression of Ihog stabilizes roughly 75% of all cytonemes by slowing down their elongation/retraction velocities, without affecting their length (*González-Méndez et al., 2017*).

We first investigated if cytonemes were normal in the absence of either Ihog or Boi as well as the requirement of Ihog and Boi for cytoneme formation and dynamics. We compared the dynamics of cytonemes after expressing GMA-GFP (*Video 1B*) or expressing GMA-GFP in the absence of Ihog and Boi in the P compartment (*Video 1C*). Cytonemes labeled with GMA-GFP, that are downregulated for Ihog and Boi, have dynamics similar to those of wild-type cells expressing only GMA-GFP (*Figure 5C*). Based on the results obtained after knocking down Ihog and Boi through RNAi expression (*Figure 5—figure supplement 1*), it appears that wild-type cells may not need Ihog and Boi to produce cytonemes, even though ectopic expression of Ihog modulates cytoneme dynamics.

On the other hand, since Boi and Ihog have been proposed to have redundant functions in Hh reception (*Zheng et al., 2010*), we explored if this redundancy also applies to the induction of cytoneme stability. We found that overexpression of Boi does not stabilize cytonemes: Hh-producing abdominal histoblasts ectopically expressing Boi emit dynamic cytonemes (*Video 2*, *Figure 5C*), while co-overexpression of Boi and Ihog leads, as predicted, to more stable cytonemes (*Video 3*). We next examined if Boi knockdown influences ectopic Ihog-driven cytoneme stabilization. To do this, we expressed throughout development Ihog-RFP and silenced Boi at the same time in the P compartment, and found that in those conditions cells showed stabilized cytonemes (*Video 4*). These results indicate that, unlike Ihog, ectopic expression of Boi does not affect cytoneme dynamics.

## The Ihog FNIII domains can modulate cytoneme dynamics

Once we had found that Ihog recruits glypicans mainly at the basolateral side of the epithelium where cytonemes are formed, we then analyzed whether the Ihog–glypican interaction is responsible for the regulation of cytoneme dynamics by testing the different Ihog domain(s) in relation to

**Table 1.** Statistical analysis of the Dally recruitment for different Ihog mutants.

p-values obtained from pairwise T test to statistically compare the Dally recruitment for different Ihog mutants (gray: n.s = not significant; orange: significant, with the corresponding p-value in scientific notation).

| Pairwise.t.test | UAS.ihogDFN-RFP | UAS.ihogDFN1-RFP | UAS.ihogDFN2-RFP | UAS.ihogFN1\*\*\*-RFP | UAS.ihogDIg-RFP |
|---|---|---|---|---|---|
| UAS.ihogDFN1-RFP | 0.027 | | | | |
| UAS.ihogDFN2-RFP | 0.023 | n.s | | | |
| UAS.ihogDIg-RFP | 4.7e-10 | 6.1e-07 | 8.1e-07 | | |
| UAS.ihogFN1\*\*\*-RFP | 1.0e-09 | 1.0e-06 | 1.3e-06 | n.s | |
| UAS.ihog-RFP | 9.9e-11 | 1.1e-07 | 1.5e-07 | n.s | n.s |

**Table 2.** Statistical analysis of the Dlp recruitment for different Ihog mutants.

p-values obtained from pairwise T test to statistically compare the Dlp recruitment for different Ihog mutants (gray: n.s = not significant; orange: significant, with the corresponding p-value in scientific notation).

| Pairwise.t.test | UAS.ihogDFN-RFP | UAS.ihogDFN1-RFP | UAS.ihogDFN2-RFP | UAS.ihogFN1***-RFP | UAS.ihogDIg-RFP |
|---|---|---|---|---|---|
| UAS.ihogDFN1-RFP | 0.0324 | | | | |
| UAS.ihogDFN2-RFP | 6.4e-06 | 0.0288 | | | |
| UAS.ihogDIg-RFP | 6.4e-07 | 0.0037 | n.s | | |
| UAS.ihogFN1***-RFP | 1.3e-06 | 0.0037 | n.s | n.s | |
| UAS.ihog-RFP | 1.8e-06 | 0.0094 | n.s | n.s | n.s |

cytoneme stabilization. Cytoneme behavior was analyzed in the Hh-producing cells measuring the lifetime of cytonemes when expressing *GMA-GFP* together with either wild-type Ihog or the different Ihog mutant variants (*Videos 5* and *6*; *Figure 5A,B*; *Figure 5—figure supplement 2B* for cytoneme dynamics graphics). All cytonemes labeled with *GMA-GFP* are also marked with Ihog-RFP constructs (*González-Méndez et al., 2017*; *Figure 5—figure supplement 2A*). As anticipated, no cytoneme stabilization was observed after ectopic expression of the Ihog intracellular fragment (*UAS.ihogCT-RFP*) (*Figure 5A*; *Video 5D*). Complementarily, 70-80% of stabilized cytonemes were observed after the ectopic expression of Ihog lacking the C-terminal fragment (*UAS.ihogΔCT-RFP*) (*Figure 5A*; *Video 5C*). Ihog cytonemes were also dynamic when co-expressing *GMA* and Ihog lacking FNIII domains (*UAS.ihogΔFn-RFP*) (*Figure 5A*; *Video 5F*). The lifetime and extent of *UAS.ihogΔFn-RFP* cytonemes visualized with both RFP and GMA-GFP were like those of wild-type cytonemes expressing GMA-GFP alone (*Figure 5—figure supplement 1*). Likewise, the absence of either the Fn1 (*Figure 5A*; *Video 5G*) or the Fn2 (*Figure 5A*; *Video 5H*) domains results in no cytoneme stabilization. However, surprisingly, the absence of the Ig domain (*UAS.ihogΔIg-RFP*) does not stabilize cytonemes (*Figure 5A*; *Video 5E*), despite the normal interaction of this construct with glypicans. Interestingly, we have also noticed that the expression levels of *UAS.ihogΔIg-RFP* at membranes are lower than those of the rest of the constructs, and this difference could give rise to this unexpected effect on cytoneme dynamics. An alternative explanation is that the Ig domains could interact with extracellular matrix components other than glypicans that could also influence cytoneme dynamics.

Since both FNIII domains (Fn1 and Fn2) affect cytoneme dynamics, we then analyzed the effect on cytoneme stabilization of the point mutation constructs. The expression of *UAS.ihogFn1***-RFP*, affecting Ihog interaction with Hh (*Figure 3E*) but not with glypicans (*Figure 3H,H'*) resulted in cytoneme stabilization (*Figure 5B*; *Video 6B*), showing that the ability of Ihog overexpression to modulate cytoneme dynamics is not dependent on the same amino acids needed for Hh binding, but rather on those involved in the interaction with glypicans. In contrast, the expression of *UAS.ihogFn2**-, described to modify the interaction of Ihog with Ptc (*Zheng et al., 2010*), does not stabilize cytonemes with or without RFP (*Figure 5B*; *Video 6C,D*). Nevertheless, this construct performs as non-functional since it does not turn out the same effects as the lack of the entire Fn2 domain

**Table 3.** Statistical analysis of the Hh recruitment for different Ihog mutants.

p-values obtained from pairwise T test to statistically compare the Hh recruitment for different Ihog mutants (gray: n.s = not significant; orange: significant, with the corresponding p-value in scientific notation).

| Pairwise.t.test | UAS.ihogDFN-RFP | UAS.ihogDFN1-RFP | UAS.ihogDFN2-RFP | UAS.ihogFN1***-RFP | UAS.ihogDIg-RFP |
|---|---|---|---|---|---|
| UAS.ihogDFN1-RFP | n.s | | | | |
| UAS.ihogDFN2-RFP | 3.3e-06 | 9.6e-06 | | | |
| UAS.ihogDIg-RFP | 0.012 | 0.027 | 0.027 | | |
| UAS.ihogFN1***-RFP | n.s | n.s | 9.6e-06 | 0.027 | |
| UAS.ihog-RFP | <2e-16 | <2e-16 | <2e-16 | <2e-16 | <2e-16 |

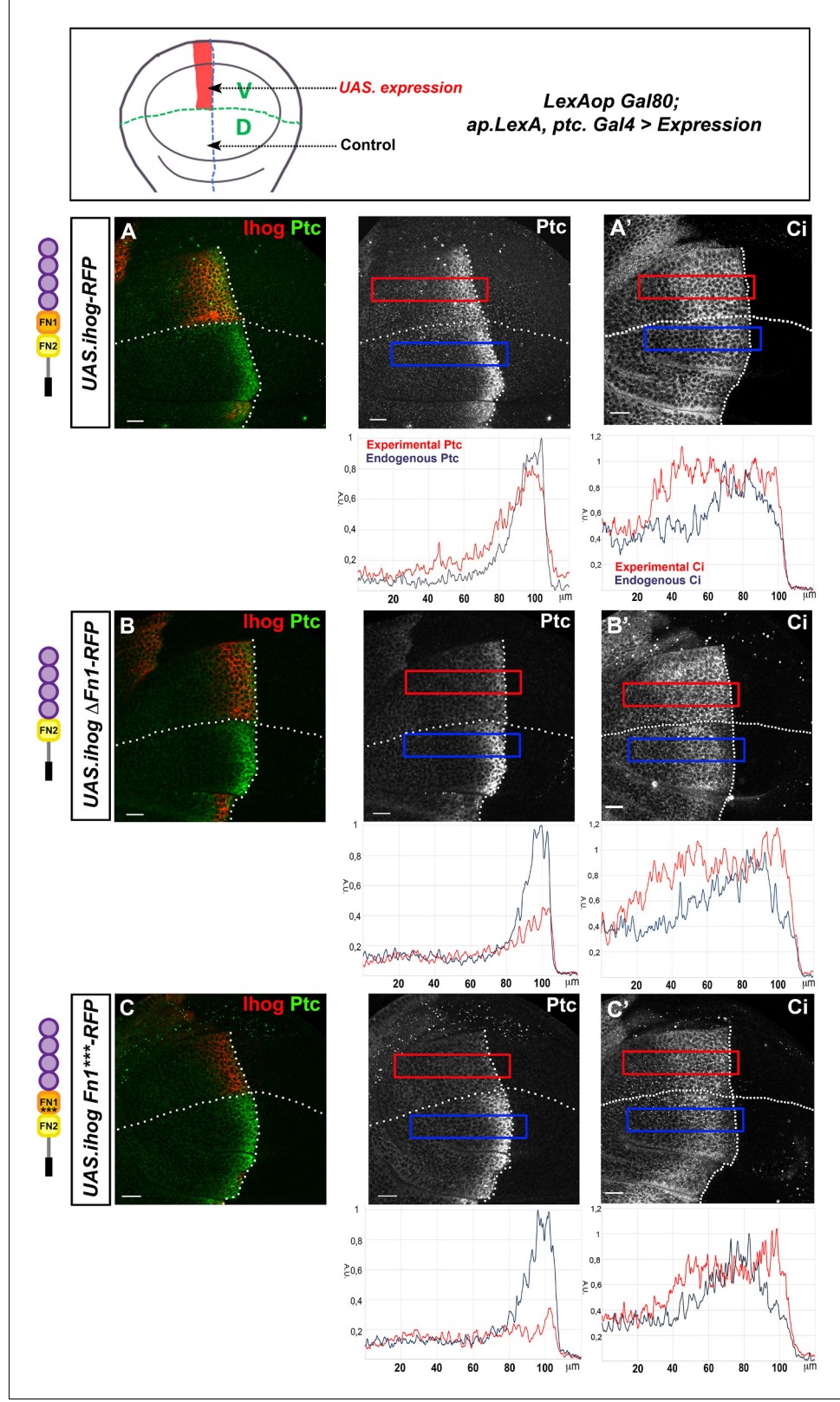

**Figure 4.** Effect of the ectopic expression of Ihog mutant forms on Hh gradient formation. (**A–C**) Ptc (**A–C**) and Ci (**A'–C'**) expression in wing discs after 28 hr induction of: *UAS.ihog-RFP* (**A**), *UAS.ihogΔFn1-RFP* (**B**), and *UAS. ihogFn1\*\*\*RFP* (**C**) using the multiple driver *LexAopGal80; apLexA ptcGal4/+; tubGal80^{ts}/+*. Images incorporate underneath plots of the fluorescence intensity of Ptc and Ci expression in the ventral experimental side (red)
*Figure 4 continued on next page*

*Figure 4 continued*

compared with the dorsal control side (blue) of the wing disc. Note that both IhogΔFn1 and IhogFn1*** strongly reduce Hh reception. Discs are oriented with dorsal (D) part down and posterior (P) right. Plot profiles were performed over a ROI of size 120 μm × 20 μm. The discs shown in panels are representative of at least five discs in three independent experiments. Scale bar: 20 μm.

The online version of this article includes the following figure supplement(s) for figure 4:

**Figure supplement 1.** Effects of the ectopic expression of IhogΔFn2-RFP and IhogFn2**-RFP in Hh signaling.

**Figure supplement 2.** Effect of the Ihog Fn1 domain on high-threshold targets of Hh.

**Figure supplement 3.** Endogenous Ihog protein decreases by the expression of IhogΔFn1 and IhogFn1***.

---

when expressing it, except for a decrease in the amount of endogenous Dally (*Figure 3—figure supplement 2*).

In summary, these results show that the two Ihog FNIII domains (which are also responsible for the Ihog–glypicans interaction) have a predominant role in cytoneme stabilization, and this stabilization is not dependent on the same residues that regulate the Ihog–Hh interaction.

## Ihog and Boi functions are not interchangeable for Hh gradient formation

The data presented so far indicate distinctive roles for Ihog and Boi in Hh gradient formation. First, as Boi has a more apical subcellular location and Ihog more basal (*Bilioni et al., 2013*; *Hsia et al., 2017*), it was not surprising to find that ectopic Boi accumulates glypicans apically, while ectopic Ihog does it basally (*Figure 1—figure supplement 1*). Second, glypicans are required to stabilize Ihog, but not Boi, at the plasma membranes (*Figure 1F,G* respectively). Third, excess of Ihog but not of Boi, modulates cytoneme dynamics (*Videos 1*, *2*, and *4*).

Thus, to explore Ihog and Boi potentially different requirements for Hh gradient formation, we generated $ihog^{-/-}$ mutant clones abutting the A/P compartment border in a wild-type background (*Figure 6A*) or in a *boi* mutant background (*Figure 6B*). Unexpectedly, the lack of Ihog is sufficient to reduce Hh responses in Hh-receiving cells despite Boi presence (*Figure 6A*, dotted line area, clone 1), while the absence of Boi does not reduce Hh responses in the same way when Ihog is present (*Figure 6B*, yellow arrowhead). In addition, the low Hh threshold target Ci is also reduced in $ihog^{-/-}$ mutant clones located in the Hh reception area but not touching the A/P compartment border (*Zheng et al., 2010*; *Figure 6A*, dotted line area, clone 2). On the other hand, as expected, the lack of both co-receptors abolishes the Hh signaling gradient (*Figure 6B*, dotted line area). These results demonstrate a non-redundant role of Ihog and Boi in Hh reception.

In agreement with previous results, the knockdown of Ihog by RNAi expression in the ventral receiving cells reduces the expression of the high (En and Ptc) and the low (Ci) threshold Hh targets, compared to the wild-type control dorsal side (*Figure 6C*) and in spite of an increase in Boi levels under these experimental conditions (*Figure 6—figure supplement 1*). We then

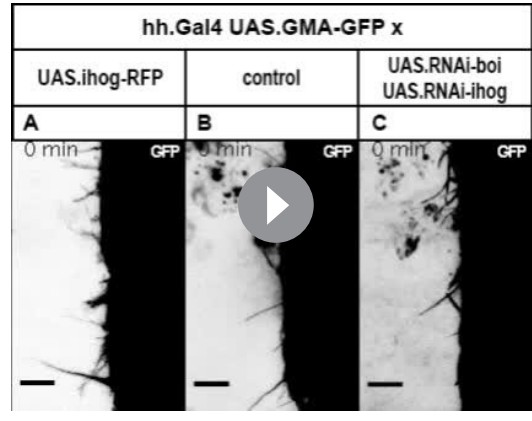

**Video 1.** Cytoneme dynamics under different levels of Ihog and Boi in histoblasts. Abdominal histoblasts of pupae expressing UAS.GMA-GFP (black) under the control of Hh.Gal4; tubGal80ts and also expressing UAS.Ihog-RFP (after 24 hr of expression) (A). Note the stabilization of cytonemes compared with the very dynamic cytonemes of the control (expression of only GMA-GFP) (B). Abdominal histoblasts of a pupa continuously expressing UAS.RNAi-Ihog and UAS.RNAi-Boi in the P compartment during development (C) show cytonemes with similar dynamics to that of the control expressing only GMA-GFP (B). Movies are 30 min long. Frames correspond to the projection of a Z-stack at 2 min intervals. The A compartment is on the left. Pupae were around 30 hr after puparium formation. Scale bar: 10 μm.

https://elifesciences.org/articles/64581#video1

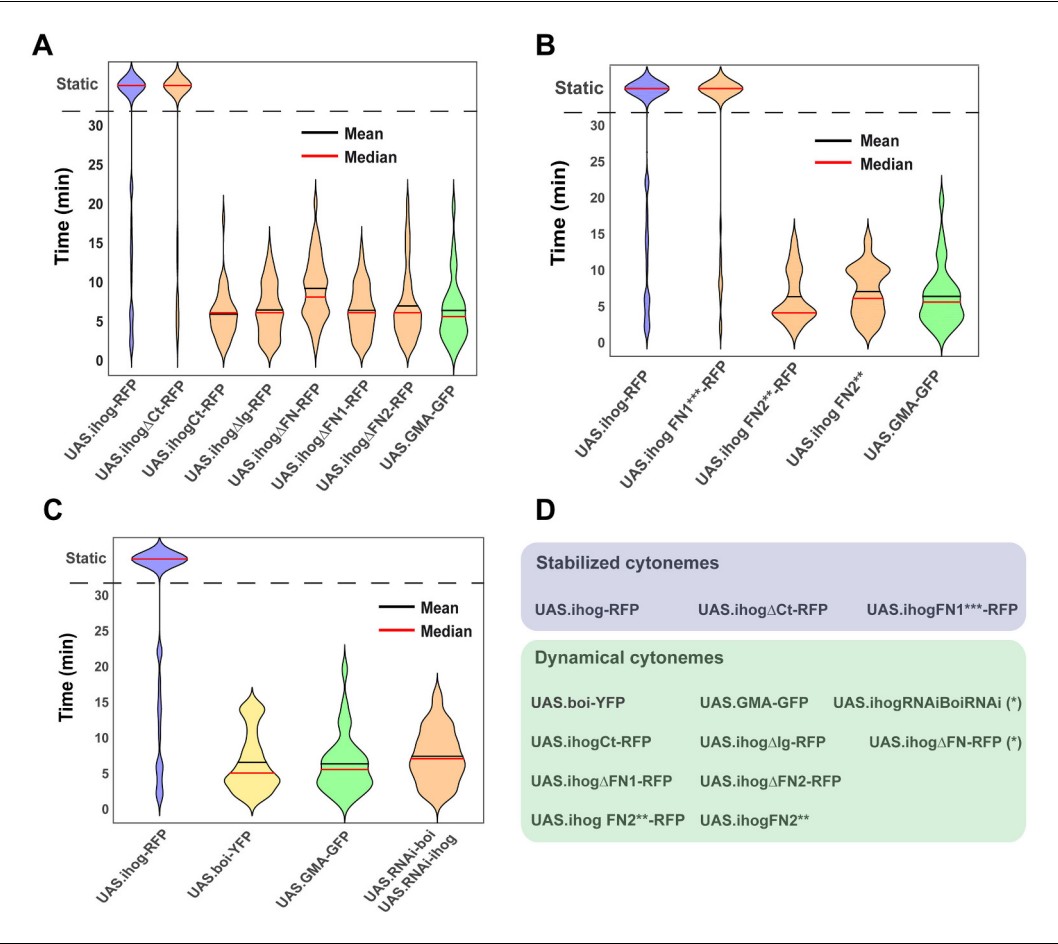

**Figure 5.** Effect of Ihog and its mutant forms in the regulation of cytoneme dynamics. (**A**) Violin plots of lifetime distributions of dynamic cytonemes in different Ihog overexpression genotypes: *UAS.ihog-RFP* (blue), wild-type control *UAS.GMA-GFP* (green), and different Ihog deletion mutants (orange). (**B**) Comparison between the effect of the ectopic expression of different Ihog point mutations: Control *UAS.ihog-RFP* (blue), wild-type control *UAS. GMA-GFP* (green); mutants of Ihog (orange). (**C**) Comparison between different levels of Ihog and Boi: *UAS.ihog-RFP* (blue), *UAS.boi-YFP* (yellow), *UAS.GMA-GFP* (green), and *UAS.RNAi-Boi/UAS.RNAi-ihog* (orange). (**D**) Table summarizing the dynamic of cytoneme in different genotypes: Stabilized cytonemes in blue and dynamical cytonemes in green. (*) Statistically significant lifetime differences compared to wild-type *GMA-GFP* cytonemes. Data is available at *Figure 5—source data 1* and p-values of the statistical analysis are shown in *Tables 4* and *5* (Material and methods).

The online version of this article includes the following source data and figure supplement(s) for figure 5:

**Source data 1.** Cytonemes lifetime measures under different experimental conditions.
**Figure supplement 1.** Ihog and Boi RNAis effects.
**Figure supplement 2.** Cytoneme dynamics after expressing the Ihog mutant constructs.

---

analyzed the effect on Hh gradient of knocking down Boi without altering Ihog. We observed the opposite effects: a slight flattening of the Hh gradient with a decrease in the high levels of Ptc, a slight extension of the Ci gradient and a maintenance of En levels in the A compartment (*Figure 6D*). These results indicate that in the absence of Ihog the long- and short-range gradient of Hh is affected and it is not rescued by the co-receptor Boi, which is only capable of partially maintain the Hh responses.

## Discussion

### Glypicans and Ihog interaction

There is strong experimental evidence for the glypicans Dally and Dlp to regulate the release, dispersal, and reception of Hh in *Drosophila* (reviewed in *Yan and Lin, 2009*; *Filmus and Capurro, 2014*). Although the glypican core region of Dlp is known to function as a Hh signaling binding protein, a direct high-affinity interaction between Dlp and either Hh or the Hh-Ihog complex had not yet been detected (*Williams et al., 2010*). On the other hand, it has previously been postulated that Ihog and Boi, two adhesion molecules that act as Hh coreceptors, are required for all known biological responses to Hh signaling during embryonic and imaginal development (*Yao et al., 2006b*; *Camp et al., 2010*; *Zheng et al., 2010*).

In this report, we show evidence of the Ihog–glypican protein interaction in wing imaginal discs. First, we demonstrated that Ihog is stabilized at the plasma membrane by glypicans, since the level of Ihog drastically decreases in double mutant clones for genes encoding glypicans (*dally* and *dlp*). Interestingly, this interaction between glypicans and Ihog could be mediated by the HS-GAG chains of the glypicans, as the same effect is observed in clones mutant for the enzymes that synthetize the HS-GAG chains (*ttv* and *btv*). The molecular mechanisms leading to this regulation are still unknown; nonetheless, as glypicans are required for Hh (*Bilioni et al., 2013*) and Wg recycling (*Gallet et al., 2008*) they might also be required for a hypothetical recycling of Ihog. Second, we showed a different role of glypicans with respect to Boi: under the same experimental conditions, Boi plasma membrane levels are not affected, indicating that the modulation of Ihog by glypicans is specific. Third, we determined that the glypicans–Ihog interaction is mediated by both Ihog FNIII domains, Fn1 and Fn2; however, even though the Fn1 domain interacts equally with both glypicans (see also *Yang et al., 2021*), the Fn2 domain is almost dispensable for the interaction with Dlp, but necessary for that with Dally.

### Ihog–glypican interaction regulates cytoneme dynamics

In wing discs and in the abdominal histoblast nests, the ectopic expression of Ihog stabilizes cytonemes, although without affecting their normal length (*Bischoff et al., 2013*; *González-Méndez et al., 2017*). On the other hand, general cytoneme establishment and dynamics are not affected after inhibiting Ihog and Boi using actin reporters to visualize them. However, with our current experimental tools we cannot discriminate Hh specific cytonemes from those

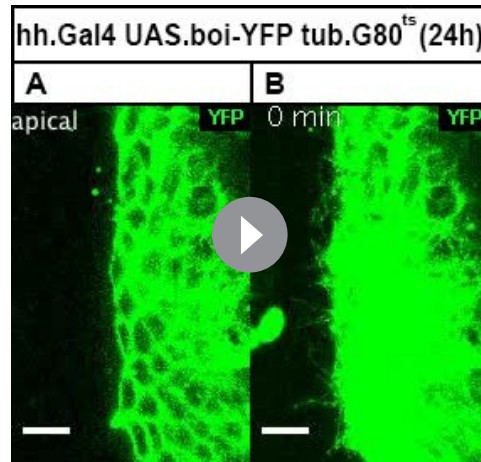

**Video 2.** Cytoneme dynamics of abdominal histoblasts expressing *UAS.boi-YFP*. Abdominal histoblasts of a *UAS.boi-YFP/+; hh.Gal4 tubGal80^{ts}* pupa expressing Boi-YFP (green) during 24 hr. (A) Z-stack from apical to basal, showing cytonemes in the most basal part. (B) 30 min movie of the same pupa showing dynamic cytonemes. Frames correspond to the projection of a Z-stack at 2 min intervals between frames. The A compartment is on the left. Pupa was around 30 hr after puparium formation. Scale bar: 10 μm.
https://elifesciences.org/articles/64581#video2

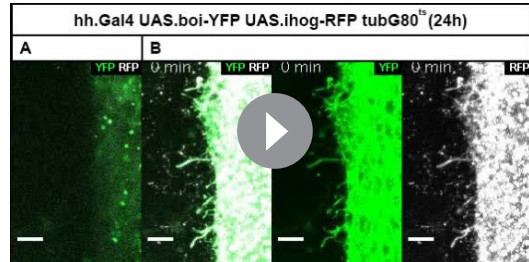

**Video 3.** Cytoneme dynamics of histoblasts expressing both *UAS.boi-YFP* and *UAS.ihog-RFP*. Abdominal histoblasts of a *UAS.boi-YFP/UAS.ihog-RFP; hh.Gal4 tubGal80^{ts}* pupa. Boi-YFP (green) and Ihog-RFP (gray) were expressed during 24 hr before recording. (A) Z-stack from apical to basal showing basal cytonemes. (B) 30 min movie of the same pupa showing stabilized cytonemes. Frames correspond to the projection of a Z-stack at 2 min intervals. The A compartment is on the left. Pupa was around 30 hr after puparium formation. Scale bar: 10 μm.
https://elifesciences.org/articles/64581#video3

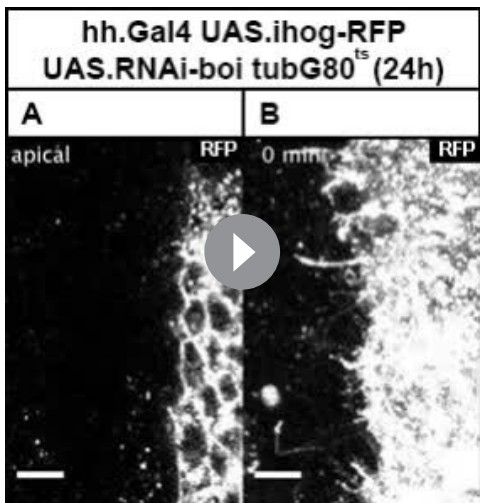

**Video 4.** Cytoneme dynamics of histoblasts expressing *UAS.ihog-RFP* and *UAS-RNAi-boi*. Abdominal histoblasts of a *UAS.Ihog-RFP/UAS.RNAi-Boi; Hh.Gal4 tubGal80ᵗˢ/+* pupa. Ihog-RFP (gray) and RNAi-Boi were expressed during 24 hr before recording. (A) Projection of a Z-stack from apical to basal show basal location of cytonemes. (B) 30 min movie of the same pupa showing stabilized cytonemes. Frames correspond to the projection of a Z-stack at 2 min intervals. The A compartment is on the left. Pupa was around 30 hr after puparium formation. Scale bar: 10 μm.
https://elifesciences.org/articles/64581#video4

domain is involved in the recruitment of Hh and the two glypicans Dally and Dlp at cytonemes. In addition, the expression of Ihog without the Fn2 domain results on dynamic cytonemes in abdominal histoblast nests. Ihog ΔFn2 lacks the ability to interact with the glypican Dally, but still retains its interaction with Dlp, suggesting a more specific role of Dally in cytoneme dynamics. Besides, the expression of Ihog lacking the intracellular domain still stabilizes cytonemes, while the expression of the intracellular domain only does not. Therefore, the intracellular fragment, potentially in contact with the actin machinery (reviewed in *Sanchez-Arrones et al., 2012*) needed for the elongation and retraction of cytonemes, does not appear to influence their stabilization.

## Ihog binding to Hh is influenced by its interaction with glypicans

Ihog has a known role in Hh recruitment to the membranes of both producing and receiving cells in the wing disc (*Yan et al., 2010*; *Bilioni et al., 2013*). It has also been proposed that Ihog may act as an exchange factor by retaining Hh on the cell surface, competing with Dlp for Hh binding (*Yan et al., 2010*). In

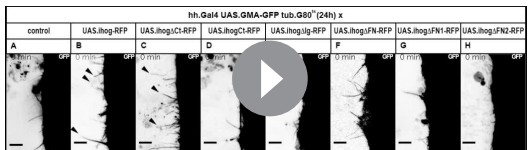

**Video 5.** Roles of Ihog domains on cytoneme stability. Abdominal histoblasts expressing *UAS.GMA-GFP* (black) under the control of *Hh.Gal4 tubGal80ᵗˢ* pupae (24 hr of expression) (A) and also expressing *UAS.Ihog-RFP* (B); *UAS.Ihog-ΔCT-RFP* (C); *UAS.IhogCT-RFP* (D); *UAS.Ihog-ΔIg-RFP* (E); *UAS.Ihog-ΔFn-RFP* (F); *UAS.Ihog-ΔFn1-RFP* (G) and *UAS.Ihog-ΔFn2-RFP* (H). Note that cytonemes are stable in (B and C), while they are dynamic in (D–H). Movies are 30 min long. Frames correspond to the projection of a Z-stack at 2 min intervals. The A compartment is on the left. Pupae were around 30 hr after puparium formation. Scale bar: 10 μm.
https://elifesciences.org/articles/64581#video5

probably used for other signaling pathways. Thus, the formation of cytonemes may probably be regulated by a general cell machinery, notwithstanding the existence of pathway specific regulation.

It was interesting finding out that the Ihog cytoneme stabilization effect depends on these extracellular FNIII domains. In particular, the Fn1

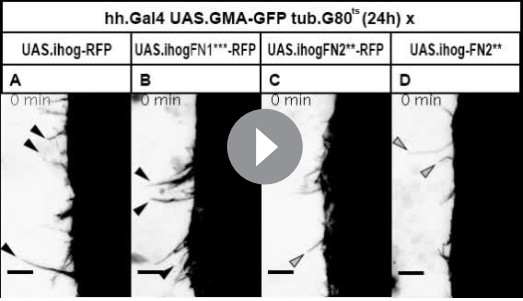

**Video 6.** Different effects of Ihog-FN1*** and IhogFN2** on cytoneme stability. Abdominal histoblasts of pupae expressing *UAS.GMA-GFP* (in black) and *UAS.ihog-RFP* (A); *UAS.ihog-Fn1\*\*\*-RFP* (B); *UAS.ihogFn2\*\*-RFP* (C); *UAS.ihogFn2\*\** (D) under the *Hh.Gal4 tubGal80ᵗˢ* control (24 hr of induction). Note that cytonemes from cells expressing Ihog-Fn1***-RFP, that alters its interaction with Hh but not with glypicans (B), are as stable as those from the control cells expressing Ihog-RFP (A), while IhogFn2** presents normal cytoneme dynamics (C, D). Movies are 30 min long. Frames correspond to the projection of a Z-stack at 2 min intervals. The A compartment is on the left. Pupae were around 30 hr after puparium formation. Scale bar: 10 μm.
https://elifesciences.org/articles/64581#video6

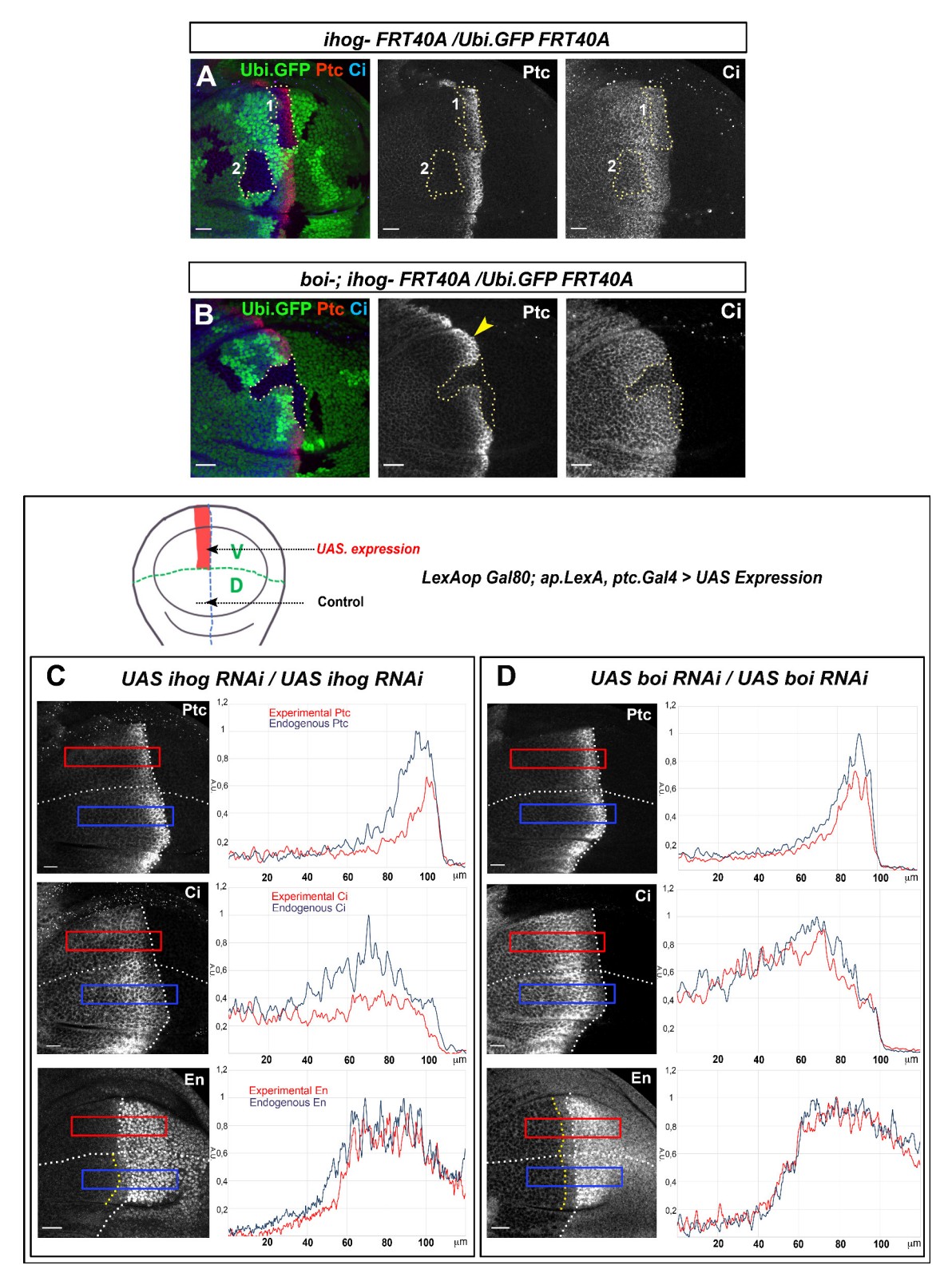

**Figure 6.** Ihog and Boi have differential roles on Hh gradient formation. (**A**) Homozygous *ihog^Z23* mutant clones (labeled by the lack of GFP) in *boi* wild-type background (**A**) and in *boi^−/−* mutant background (**B**). Both experiments are immunostained for Ptc and Ci. Ptc expression is significantly reduced in *ihog^Z23* mutant clones located in the A compartment abutting the A/P compartment border (clone one in **A**) and Ci is reduced in clones close to the A/P border in the Hh signaling zone (clone two in **A**). However, Ptc and Ci expressions are maintained in a *boi^−* mutant background (**B**, yellow

*Figure 6 continued on next page*

*Figure 6 continued*

arrowhead). Note that there is not Hh signaling in the absence of both Ihog and Boi (B, dotted line area). (C, D) Ptc, Ci, and En expression in wing discs after 48 hr induction of *UAS.ihog-RNAi* (C) and *UAS.boi-RNAi* (D) using the multiple driver *LexAopGal80; apLexA ptcGal4/+; tubGal80$^{ts}$/+* (the scheme above shows the ventral/anterior domain of induction using this multiple driver). The *UAS.ihog-RNAi* expression reduces Ptc, Ci, and En levels (C), while extends Hh gradient slightly (Ptc and Ci show a more flattened pattern) after *UAS.boi-RNAi* expression (D). Discs are oriented with dorsal (D) part down and posterior (P) right. Plot profiles were performed over a ROI of size 120 µm × 20 µm for Ptc, Ci and En. The discs shown in panels are representative of at least five discs in three independent experiments. Scale bar: 20 µm.

The online version of this article includes the following figure supplement(s) for figure 6:

**Figure supplement 1.** The lack of Ihog result in a Boi increase.

addition, of the three Ihog extracellular domains (Fn1, Fn2, and Ig), only Fn1 and Fn2 are necessary for in vivo Hh signaling, although accomplishing independent functions (*McLellan et al., 2006*; *Zheng et al., 2010*).

The Fn1 domain has been described as specifically needed for Hh binding (*McLellan et al., 2006*; *Yao et al., 2006b*). In agreement, we have observed that the ectopic expression of the variant of Ihog lacking Fn1 does not accumulate Hh in the P compartment as the ectopic wild-type Ihog does. Specifically, the amino-acid changes (D558N, N559S, E561Q) that we have generated in the Fn1 domain (Ihog-Fn1***) impede Hh accumulation. These mutations are not identical to those described by *McLellan et al., 2006*, although they map in the same Ihog protein region predicted to interact with Hh. Interestingly, our amino-acid alterations maintain the Ihog–glypican interaction, showing that both interactions (with Hh and with glypicans) are mediated by different sequences within the same Ihog extracellular domain. Accordingly, crystal structure of the Hh-binding fragment of Ihog has shown that the Fn1 domain is specifically needed for heparin binding-dependent Ihog dimerization as well as required for high-affinity interactions between Ihog and Hh. Also in agreement with our data, the putative heparin-binding site near the Ihog–Ihog dimer interface was found to be separated from the basic strip bridging Hh to Ihog-Fn1 (*McLellan et al., 2006*). Besides, we have also observed that ectopic expression of a construct carrying only the wild-type Fn1 domain recruits low levels of Hh, suggesting that, although Fn1 binds physically to Hh, cooperation between Fn1 and Fn2 domains is needed for an efficient Fn1-Hh binding. Both domains are required for proper Ihog–Dally interaction, what explains why the Fn2 domain also slightly influences Hh accumulation.

## Ihog, but not Boi, function is needed for cytoneme-mediated Hh gradient

As Hh co-receptors, the functions of Ihog and Boi have been described to be interchangeable for Hh signaling (*Yao et al., 2006b*; *Camp et al., 2010*; *Zheng et al., 2010*). Nevertheless, here we have shown different functional requirements for each protein: (1) ectopic Ihog, but not Boi, accumulates glypicans in the basal part of the wing disc tissue; (2) the levels of glypicans are key to maintain Ihog, but not Boi, protein levels at cytoneme membranes; and (3) Ihog, but not Boi, has a cytoneme-stabilizing function in vivo. These observations overall suggest a specific function of Ihog and its interaction with glypicans to stabilize cytonemes and regulate the Hh gradient. In agreement, we show that Hh gradient was affected in the absence of Ihog, even though Boi levels are maintained. Although Ihog and Boi are both part of the Ptc-coreceptor complex, our data demonstrate that Boi cannot substitute Ihog function in the formation of the Hh gradient (*Figure 7B*). We have previously proposed that cytonemes at the basal part of the epithelia regulate both the short- and long-range Hh gradient, while apical contacts significantly contribute to the short-range gradient (*Callejo et al., 2011*; *Bilioni et al., 2013*; *Chen et al., 2017*; *González-Méndez et al., 2017*). Accordingly, basal glypicans–Ihog interactions indicate a main function of Ihog in long-range Hh gradient formation. Boi is located more apically in the disc epithelium (*Bilioni et al., 2013*; *Hsia et al., 2017*), where no long cytonemes have been observed (*Bilioni et al., 2013*). Besides, the apical reception complex could have an important input in Hh high-threshold or short-range responses (*Callejo et al., 2011*).

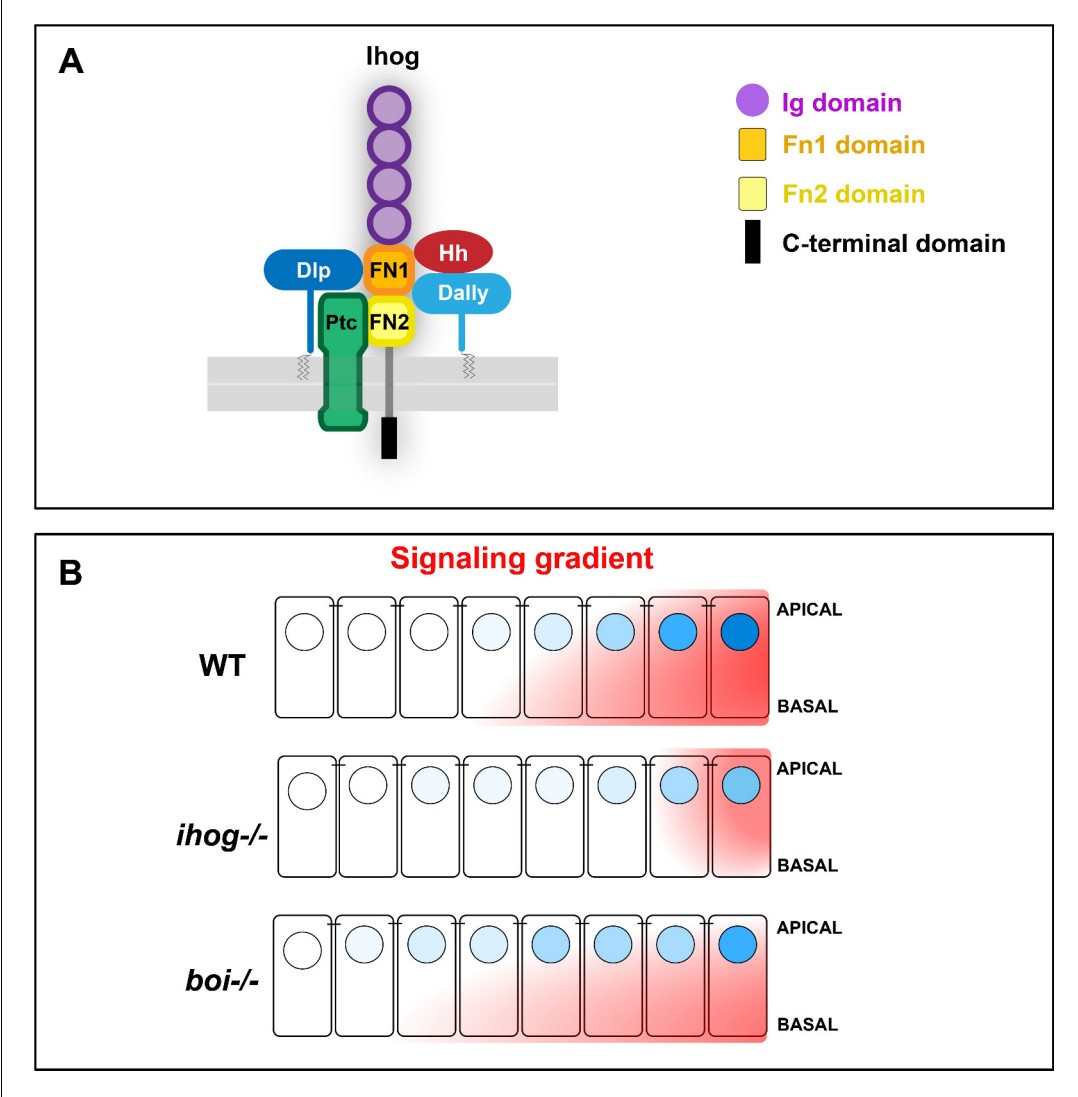

**Figure 7.** Scheme representatives of glypicans, Hh, and Ptc interaction with Ihog and its specific requirement for long-range Hh gradient. (**A**) Scheme depicting the Ihog domains and their ineractions of the two FNIII domains with Hh (*Yao et al., 2006b*), Ptc (*Zheng et al., 2010*), and the glypicans Dally and Dlp. (**B**) Ihog and Boi loss of function differentially affect the formation of the Hh signaling gradient: while Ihog seems to be required for the short- and long range, boi is needed for the short-range Hh gradient.

The online version of this article includes the following figure supplement(s) for figure 7:

**Figure supplement 1.** Summary of the effects on Dally, Dlp, Hh gradient, and cytoneme dynamics after ectopic expression of wild-type Ihog and its mutant forms.

## The ability of glypicans to regulate Hh signaling depends on their interaction with Ihog at cytonemes

Hh-graded distribution across the receiving *Drosophila* epithelia is mediated by cytonemes and reception seems to happen at discrete contact sites along presenting and receiving cytonemes. At these contact places, most of the extracellular components needed for Hh signaling colocalize in a dynamic way (*González-Méndez et al., 2017*), similar to a synaptic process (*Huang et al., 2019*; *González-Méndez et al., 2020*). Our results support the hypothesis that the interactions of Hh, Ptc, and Ihog with the glypicans, among other complex components, might guide the Hh reception at cytoneme contact sites. In this context, it has recently been proposed that the Ihog–Ihog homophilic interaction is competing with the Ihog-Hh heterophilic interaction through the Ihog Fn1 domain (*Yang et al., 2021*). This competing interaction at the cytoneme contact sites between presenting

and receiving cytonemes could allow the Hh reception. On the other hand, recent research describes that the Dlp class of glypicans shields the highly hydrophobic lipid moiety of Wnt from the aqueous environment, allowing its handing over to receptors (*McGough et al., 2020*). For Hh signaling, Dlp could also tunnel Hh lipid modifications while transferring Hh to its receptor Ptc and coreceptors Ihog and Boi. Interestingly, Dlp, as part of the reception complex, interacts specifically with the Ihog Fn1 domain, the same domain that binds Hh.

Glypicans control several morphogenetic gradients (Bmp, FGF; Dpp; Jak/Stat) in addition to those of Hh and Wnt. Based on our data, glypican regulation over Hh signaling is mainly given by their interaction with Ihog, thus conferring pathway specificity. We, therefore, propose that the glypicans–Ihog interaction could aid cytoneme dynamics, Hh-maintenance at cytonemes, Hh presentation to Ptc, as well as Hh release during the reception process.

# Materials and methods

**Key resources table**

| Reagent type (species) or resource | Designation | Source or reference | Identifiers | Additional information |
|---|---|---|---|---|
| Transgenic construct (*Drosophila melanogaster*) | *UAS.GMA-GFP* | *Bloor and Kiehart, 2001* | | |
| Transgenic construct (*Drosophila melanogaster*) | *UAS.ihog-RFP* | *Callejo et al., 2011* | | |
| Transgenic construct (*Drosophila melanogaster*) | *UAS.ihog-CFP* | *Bilioni et al., 2013* | | |
| Transgenic construct (*Drosophila melanogaster*) | *UAS.ihogΔCT-RFP* | Available in Isabel Guerrero's laboratory, Severo Ochoa Molecular Biology Center, Madrid, Spain | | |
| Transgenic construct (*Drosophila melanogaster*) | *UAS.ihogΔFN-RFP* | Available in Isabel Guerrero's laboratory, Severo Ochoa Molecular Biology Center, Madrid, Spain | | |
| Transgenic construct (*Drosophila melanogaster*) | *UAS.ihogΔIg-RFP* | Available in Isabel Guerrero's laboratory, Severo Ochoa Molecular Biology Center, Madrid, Spain | | |
| Transfgenic construct (*Drosophila melanogaster*) | *UAS.ihogΔFn1-RFP* | Available in Isabel Guerrero's laboratory, Severo Ochoa Molecular Biology Center, Madrid, Spain | | |
| Transgenic construct (*Drosophila melanogaster*) | *UAS.ihogΔFn2-RFP* | Available in Isabel Guerrero's laboratory, Severo Ochoa Molecular Biology Center, Madrid, Spain | | |
| Transgenic construct (*Drosophila melanogaster*) | *UAS.ihogFn1***-RFP* | Available in Isabel Guerrero's laboratory, Severo Ochoa Molecular Biology Center, Madrid, Spain | | |
| Transgenic construct (*Drosophila melanogaster*) | *UAS.ihogFn2**-RFP* | Available in Isabel Guerrero's laboratory, Severo Ochoa Molecular Biology Center, Madrid, Spain | | |
| Transgenic construct (*Drosophila melanogaster*) | *UAS.ihogFn2*** | *Zheng et al., 2010* | | |

*Continued on next page*

Continued

| Reagent type (species) or resource | Designation | Source or reference | Identifiers | Additional information |
|---|---|---|---|---|
| Transgenic construct (*Drosophila melanogaster*) | *UAS.ihogCT-RFP* | Available in Isabel Guerrero's laboratory, Severo Ochoa Molecular Biology Center, Madrid, Spain | | |
| Transgenic construct (*Drosophila melanogaster*) | *UAS.CD8.Fn1-mCherry* | Available in Isabel Guerrero's laboratory, Severo Ochoa Molecular Biology Center, Madrid, Spain | | |
| Transgenic construct (*Drosophila melanogaster*) | *UAS.boi-YFP* | **Bilioni et al., 2013** | | |
| Transgenic construct (*Drosophila melanogaster*) | *UAS.ihog-RNAi* | Vienna *Drosophila* Resource Center (VDRC), ref: 102602 | | |
| Transgenic construct (*Drosophila melanogaster*) | *UAS.boi-RNAi* | Vienna *Drosophila* Resource Center (VDRC), ref: 108265 | | |
| Transgenic construct (*Drosophila melanogaster*) | *UAS.dlp-RNAi* | Vienna *Drosophila* Resource Center (VDRC), ref: 10299 | | |
| Transgenic construct (*Drosophila melanogaster*) | *UAS.dally-RNAi* | Vienna *Drosophila* Resource Center (VDRC), ref: 14136 | | |
| Genetic reagent (*Drosophila melanogaster*) | *hs-FLP* | **Golic and Lindquist, 1989** | | |
| Genetic reagent (*Drosophila melanogaster*) | *dally$^{32}$* | **Franch-Marro et al., 2005** | | |
| Genetic reagent (*Drosophila melanogaster*) | *dlp$^{20}$* | **Franch-Marro et al., 2005** | | |
| Genetic reagent (*Drosophila melanogaster*) | *ttv$^{524}$* | **Takei et al., 2004** | | |
| Genetic reagent (*Drosophila melanogaster*) | *botv$^{510}$* | **Takei et al., 2004** | | |
| Genetic reagent (*Drosophila melanogaster*) | *boi* | **Zheng et al., 2010** | | |
| Genetic reagent (*Drosophila melanogaster*) | *ihog$^{Z23}$* | **Zheng et al., 2010** | | |
| Genetic reagent (*Drosophila melanogaster*) | *Bac Hh:GFP* | **Chen et al., 2017** | | |
| Genetic reagent (*Drosophila melanogaster*) | *Bac ihog:GFP* | **Hsia et al., 2017** | | |
| Genetic reagent (*Drosophila melanogaster*) | *dally-trap-YFP* | **Lowe et al., 2014** | | |

*Continued*

| Reagent type (species) or resource | Designation | Source or reference | Identifiers | Additional information |
|---|---|---|---|---|
| Genetic reagent (*Drosophila melanogaster*) | hh.Gal4 | *Tanimoto et al., 2000* | | |
| Genetic reagent (*Drosophila melanogaster*) | ptc.Gal4 | *Hinz et al., 1994* | | |
| Genetic reagent (*Drosophila melanogaster*) | ap.Gal4 | *Calleja et al., 1996* | | |
| Genetic reagent (*Drosophila melanogaster*) | ap.LexA | Bloomington Stock Center ref:54268 | | |
| Genetic reagent (*Drosophila melanogaster*) | LexO.TubGal80 | Bloomington Stock Center ref:32217 | | |
| Genetic reagent (*Drosophila melanogaster*) | act>y+>Gal4 | *Pignoni and Zipursky, 1997* | | |
| Antibody | Mouse monoclonal α-Ptc | *Capdevila and Guerrero, 1994* | | 1:30 |
| Antibody | Mouse monoclonal α-Dlp | *Lum et al., 2003b* | | 1:30 |
| Antibody | Rabbit polyclonal α-GFP | Molecular Probes | | 1:1000 |
| Antibody | Rabbit polyclonal α-Ihog | *Bilioni et al., 2013* | | 1:100 |
| Antibody | Mouse monoclonal α-En | *Patel et al., 1989* | | 1:100 |
| Antibody | Rabbit polyclonal α-Boi | *Bilioni et al., 2013* | | 1:30 |
| Antibody | Rat monoclonal α-Ci | *Motzny and Holmgren, 1995* | | 1:20 |
| Antibody | Rabbit polyclonal α-RFP | Chromoteck | | 1:5000 WB |
| Antibody | 680RD fluorescent α-rabbit | Li-Cor | | 1:10,000 WB |

## Generation of the Ihog constructs

IhogΔCT was obtained by PCR amplification using the Ihog cDNA as a template (5'CACCA TGGACGCTGCTCACATCCTC3'; 5'CTTGTTTGGATTGTTTCCCCGGCTTC3'). The resulting product was then cloned into the entry vector pENTR/D-TOPO by directional TOPO cloning (Gateway system; Invitrogen) and introduced by recombination into the destination vector pTWR (pUAST-RFP).

IhogΔFn construct was obtained by PCR amplification using the Ihog/pENTR vector as a template (5'GAATTTAGTGCCCTTAAGCAGG3'; 5'CTGCTTCTGCCTGGAACCCT3'). The resulting product was then ligated obtaining the IhogDFn/pENTR vector, lacking the FN domains, that was then introduced by recombination into the destination vector pTWR (pUAST-RFP).

To generate the IhogΔIg, IhogΔFn1, IhogΔFn2, and the IhogCT constructs two different PCR were performed (primers), introducing a restriction site (BamHI, NdeI, or KpnI). After restriction and subsequent ligation, the product lacking the domain of interest was cloned into the entry vector pENTR/D-TOPO by directional TOPO cloning (Gateway system; Invitrogen) and introduced by recombination into the destination vector pTWR (pUAST-RFP).

Primers:

**Table 4.** Statistical analysis of the lifetime for different Ihog levels.

p-values obtained from Wilcoxon rank sum test to statistically compare cytoneme lifetimes for different levels of Ihog (n.s, not significant, in gray; significant with the corresponding p-value in scientific notation, in orange).

| Wilcoxon rank sum test | UAS.GMA-GFP | UAS.RNAi-boi UAS.RNAi-ihog |
|---|---|---|
| UAS.boi-YFP | n.s | n.s |
| UAS.GMA-GFP | | 5.00E-02 |
| UAS.RNAi-boi;UAS.RNAi-ihog | | |

ΔIG PCR1: 5'CACCATGGACGCTGCTCACATCCTC3';5'CACCAAAGATTCGGATCCGCGAAG 3'
PCR2: 5'CAGATTCAGGGATCCCGGGAATCG3';5'CACGCCAACGCTGTTGAGGCT3'
ΔFn1 PCR1: 5'CACCATGGACGCTGCTCACATCCTC3';5'CTTCTGCCTGGTACCCTGA TTGGGC3'
PCR2:5'GCAGCCAGGTACCGCACTTGATCCG3';5'CACGCCAACGCTGTTGAGGCT3'
ΔFn2 PCR1:5'CACCATGGACGCTGCTCACATCCTC3';5'GTATTCCTCGATCTCCATATGCTC TGGCAC3'
PCR2:5'GAATTTAGTGCCCATATGCAGGGG3';5'CACGCCAACGCTGTTGAGGCT3'
CT PCR1:5'CACCATGGACGCTGCTCACATCCTC3'; 5'CACCAAAGATTCGGATCCGCGAAG3'
PCR2:5'CGCACCCAAGGATCCAAAACCAGC3';5'CACGCCAACGCTGTTGAGGCT3'

To generate the construct IhogFN1***, carrying point mutations D558N, N559S, and E561Q, a PCR was performed with the following primers: 5'TGAAGTTGGAGTGTAAGGCCA3' and 5'TCAAG TTCAACGTCAGGAAGTCA3'. pTWR-IhogRFP vector was digested using BglII and XhoI restriction enzymes. The PCR product and the digested vector were put together by Gibson Assembly.

The construct IhogFN2** (point mutations K653E and Q655E) was generated using pTWR-IhogRFP as a template. First, a mutagenized fragment was generated by exchanging the first nucleotide for guanosine in the codons of K653 and Q655 (*Zheng et al., 2010*). The mutagenized fragment was then inserted into pTWR-IhogRFP by digestion with Xho1 and NgoMIV.

The construct Fn1-CD8mCherry was generated by amplification of FN1 domain with the primers: 5'AGGCATGCATCAAGTGCTGCACCTG3' and 5'ACGGTACCCCACCGACTCCTCCAAATG3'. The fragment was then cloned by restriction ligation into the Kpn1 Sph1 sites of the pLOT-CD8-mCherry vector (*Harmansa et al., 2015*).

### *Drosophila* strains

The description of mutations, insertions, and transgenes is available at Fly Base (http://flybase.org). The following mutants and transgenic strains were used: *tub.Gal80^ts^*, *hs-Flp122* (Bloomington *Drosophila* Stock Centre (BDSC), Indiana, USA; http://flystocks.bio.indiana.edu), *hs-FLP* (*Golic and Lindquist, 1989*), *dally[32]* (*Franch-Marro et al., 2005*), *dlp[20]* (*Franch-Marro et al., 2005*), *ttv[524]* (*Takei et al., 2004*), *botv[510]* (*Takei et al., 2004*), *boi* (*Zheng et al., 2010*), and *ihog[Z23]* (*Zheng et al.,*

**Table 5.** Statistical analysis of the lifetime for different Ihog mutants.

p-values obtained from Wilcoxon rank sum test to statistically compare cytoneme lifetimes for different Ihog mutants with the control (UAS.GMA-GFP) (gray: n.s = not significant; orange: significant, with the corresponding p-value in scientific notation).

| Wilcoxon rank sum test | UAS.ihogCt-RFP | UAS.ihogDIg-RFP | UAS.ihogDFN-RFP | UAS.ihogDFN1-RFP | UAS.ihogDFN2-RFP | UAS.GMA-GFP |
|---|---|---|---|---|---|---|
| UAS.ihogCt-RFP | | n.s | 7.23E-06 | n.s | n.s | n.s |
| UAS.ihogDIg-RFP | | | 6.06E-04 | n.s | n.s | n.s |
| UAS.ihogDFN-RFP | | | | | 9.30E-04 | 2.05E-05 |
| UAS.ihogDFN1-RFP | | | | | n.s | n.s |
| UAS.ihogDFN2-RFP | | | | | | n.s |
| UAS.GMA-GFP | | | | | | |

*2010*). The reporter lines used were as follows: *Bac Hh:GFP* (*Chen et al., 2017*), *Bac ihog:GFP* (*Hsia et al., 2017*), and *dally-trap-YFP* (*Lowe et al., 2014*).

## Overexpression experiment

The following Gal4 and LexA drivers were used for ectopic expression experiments using Gal4/UAS (*Brand and Perrimon, 1993*) and LexA/LexAop (*Yagi et al., 2010*) systems: *hh.Gal4* (*Tanimoto et al., 2000*), *ptc.Gal4* (*Hinz et al., 1994*), *ap.Gal4* (*Calleja et al., 1996*), and *ap.LexA*. The transgene *act>y+>Gal4* (*Pignoni and Zipursky, 1997*) was used to generate random ectopic clones of the UAS lines. Larvae of the corresponding genotypes were incubated at 37°C for 12 min to induce hs-Flp-mediated recombinant clones. We use the tub-Gal80$^{ts}$ for the transient expression of transgenic constructs with the UAS/Gal4 system. Fly crosses were maintained at 17°C and dissected after inactivation of the Gal80$^{ts}$ repressor for 24 hr at restrictive temperature (29°C).

The UAS-transgenes used were as follows: *UAS.GMA-GFP* (*Bloor and Kiehart, 2001*), *UAS.ihog-RFP* (*Callejo et al., 2011*), *UAS.ihog-CFP* (*Bilioni et al., 2013*), *UAS.ihogΔCT-RFP*, *UAS.ihogΔFN-RFP*, *UAS.ihogΔIg-RFP*, *UAS.ihogΔFn1-RFP*, *UAS.ihogΔFn2-RFP*, *UAS.ihogFn1\*\*\*-RFP*, *UAS.ihogFn2\*\*-RFP*, *UAS.ihogCT-RFP*, *UAS.boi-YFP* (*Bilioni et al., 2013*), *UAS.ihog-RNAi* (VDRC 102602), *UAS.boi-RNAi* (VDRC 108265), *UAS.dlp-RNAi* (VDRC 10299), and *UAS.dally-RNAi* (VDRC 14136). *UAS.ihogFn2\*\** (*Zheng et al., 2010*), referred as (*UAS.ihogFN2\** in their study) were also used as controls to test whether or not they gave the same results as the expression of the same Ihog RFP-tagged forms and certainly they gave the same phenotypes in all the experiments.

## Clonal analysis

Mutant clones were generated by heat shock Flp-mediated mitotic recombination.

For wing imaginal disc dissection, individuals grown at 17°C were incubated at 37°C for 45 min 48–72 hr after egg laying (AEL). The genotypes were as follows:

- *y w* hsFlp$^{122}$/BacihogGFP; *dally$^{32}$ dlp$^{20}$* FRT2A / Ubi.RFP FRT2A.
- *y w* hsFlp$^{122}$/BacihogGFP; *dally$^{32}$* FRT2A / Ubi.GFP FRT2A.
- *y w* hsFlp$^{122}$/BacihogGFP; *dlp$^{20}$* FRT2A / Ubi.GFP FRT2A.
- *y w* hsFlp$^{122}$, *boi*/Y; *ihog$^{Z23}$* FRT40A / Ubi.RFP FRT40.
- *y w* hsFlp$^{122}$, *boi*/Y; *ihog$^{Z23}$* FRT 40A / Ubi.RFP FRT40; *dally trap*-YFP.

## Immunostaining of imaginal discs

Immunostaining was performed according to standard protocols (*Capdevila and Guerrero, 1994*). Imaginal discs from third-instar larvae were fixed in 4% paraformaldehyde in PBS for 20 min at room temperature (RT) and permeabilized in PBS 0.1% Triton (PBT) before incubating with PBT 1% BSA for blocking (1 hr at RT) and then with the primary antibody (overnight at 4°C). Three washes at RT for 15 min and incubation with the appropriate fluorescent secondary antibodies (ThermoFisher) at a 1:400 dilution for 1 hr at room temperature and then washing and mounting in mounting media (Vectashield). Primary antibodies were used at the following dilutions: mouse monoclonal α-Ptc (*Capdevila and Guerrero, 1994*; Hybridoma Bank Iowa), 1:30; mouse monoclonal α-Dlp (*Lum et al., 2003b*; Hybridoma Bank, Iowa), 1:30; rabbit polyclonal α-GFP (Molecular Probes, α−6455), 1/1000; rabbit polyclonal α-Ihog (*Bilioni et al., 2013*), 1:100; mouse monoclonal α-En (*Patel et al., 1989*), 1:100; rabbit polyclonal α-Boi (*Bilioni et al., 2013*), 1:30; and rat monoclonal α-Ci (*Motzny and Holmgren, 1995*): a gift from (B. Holmgren) 1:20.

## Microscopy and image processing of wing imaginal discs

Laser scanning confocal microscope (LSM710 Zeiss) was used for confocal fluorescence imaging of wing imaginal discs. ImageJ software (National Institutes of Health) was used for image processing and for image analysis.

Profiles of Ihog, Boi, Dlp, Dally, and Hh were made taking intensity gray values of a dorso-ventral region of the wing disc and normalizing to the mean gray value of the ventral compartment.

Profiles of Ptc and Ci were obtained taking intensity gray values of an anterior–posterior region in both ventral (experimental data) and dorsal compartment (control data). For the normalization of each profile, we subtracted the background using the average value of the posterior compartment and then normalized the data to the minimum value of the anterior compartment.

## In vivo imaging of pupal abdominal histoblast nests

Imaging of abdominal histoblasts of 30 hr after puparium formation pupae was done using a chamber as described in *Seijo-Barandiarán et al., 2015*. Hh signaling filopodia from histoblasts of dorsal abdominal segment A2 were filmed using 40x magnification taking Z-stacks of around 30 µm of thickness with a step size of 1 µm every 1 or 2 min, depending on the experiment, using a LSM710 confocal microscope. All movies were analyzed with Fiji. All imaged pupae developed and hatched normally.

## Quantification method and numerical analysis of glypicans and Hh recruitment

To statistically compare the glypicans and Hh recruitment under the different Ihog constructs, we measured the fluorescence intensity in the dorsal compartment of the disc and normalizing to the value of intensity in the ventral compartment (internal control). For Dally analysis, we measured six to seven discs overexpressing the following constructs (data is available at *Figure 2—source data 1*): *UAS.ihog-RFP*, n=7; *UAS.ihogΔFn-RFP*, n=6; *UAS.ihogΔFn1-RFP*, n=7; *UAS.ihogΔFn2-RFP*, n=7; and *UAS.ihogFn1\*\*\*-RFP*, n=6. For Dlp analysis, we measured five to seven discs overexpressing the following constructs (data is available at *Figure 2—source data 2*): *UAS.ihog-RFP*, n=7; *UAS.ihogΔFn-RFP*, n=7; *UAS.ihogΔFn1-RFP*, n=7; *UAS.ihogΔFn2-RFP*, n=7; and *UAS.ihogFn1\*\*\*-RFP*, n=5. For Hh analysis, we measure six to seven discs overexpressing the following constructs (data is available at *Figure 3—source data 1*): *UAS.ihog-RFP*, n=7; *UAS.ihogΔFn-RFP*, n=7; *UAS.ihogΔFn1-RFP*, n=7; *UAS.ihogΔFn2-RFP*, n=6; *UAS.ihogFn1\*\*\*-RFP*, n=7; and *UAS.cd8.Fn1-Cherry*, n=7.

Next, we performed a statistical analysis of the normality of the data employing Shapiro test in the free software R-Studio to analyze the differences between means using the T-test for normal data and the Wilcox test for non-normal data. The R-Studio code is available at *Source code 1*. p-values of the analysis are shown in the next three tables.

## Quantification method and numerical analysis of cytoneme dynamics

To statistically compare the cytoneme lifetime when *boi* and *ihog* expression is altered, we used the method described in *González-Méndez et al., 2017*, tracking cytonemes with the GMA-GFP signal (*hh.Gal4 tubG80ts UAS.ihog-RFP UAS.GMA-GFP* n=4; *hh.Gal4 UAS.GMA-GFP* n=6; *hh.Gal4 UAS.Boi-YFP/UAS.GMA-GFP* n=2; *hh.Gal4 UAS.boi-RNAi UAS.ihog.RNAi UAS.GMA-GFP* n=7). Data is available at *Figure 5—source data 1*.

Cytonemes were tracked using MTrackJ plugin of ImageJ (https://imagescience.org/meijering/software/mtrackj/). We took the base (track#1) and the tip (track#2) point coordinates of each cytoneme (cluster) and colored them by cluster, so that each cytoneme would be in a different color. The tracking was done in a region of 49 µm x 76 µm for GMA cytoneme dynamics.

To statistically compare the cytoneme lifetime when the Ihog proteins, either the wild-type or the mutant forms, are expressed ectopically, we manually quantified the frames in which cytonemes had been observed tracking the GMA-GFP signal. For each condition, we scored three to five pupae and between 10 and 20 cytonemes per pupa (*hh.Gal4 tubG80ts UAS.ihogΔCt-RFP/UAS.GMA-GFP* n=3; *hh.Gal4 tubG80ts UAS.ihogCt-RFP/UAS.GMA-GFP* n=4; *hh.Gal4 tubG80ts UAS.ihogΔIg-RFP/UAS.GMA-GFP* n=5; *hh.Gal4 tubG80ts UAS.ihogΔFn-RFP/UAS.GMA-GFP* n=5; *hh.Gal4 tubG80ts UAS.ihogΔFN1-RFP/ UAS.GMA-GFP* n=6; *hh.Gal4 tubG80ts UAS.ihogΔFN2-RFP/UAS.GMA-GFP* n=4; *hh.Gal4 tubG80ts UAS.ihogFN1\*\*\*-RFP/UAS.GMA-GFP* n=4; *hh.Gal4 tubG80ts UAS.ihogFN2\*\*-RFP/ UAS.GMA-GFP* n=5; *hh.Gal4 tubG80ts UAS.ihogFN2\*\*/UAS.GMA-GFP* n=5). Data is available at *Figure 5—source data 1*.

The manual quantification of the data was stored in an excel file and uploaded in a Matlab script. The Matlab script was designed to organize the data, to compute the statistical analysis, and to represent the results in different violin plots (*Figure 5*), and is available at *Source code 2*.

For genotypes showing only dynamic cytonemes, the statistical analysis was done using their numerical lifetimes. To examine the normality of the data distribution, we performed a Shapiro–Wilk test. Since the results showed a non-parametric distribution of the experimental data, we selected a Wilcoxon rank sum test to compare the numerical lifetimes between two genotypes. The resulting p-values can be seen in *Tables 4* and *5*.

For genotypes showing both static and non-static cytonemes, we defined a no numerical case to quantify the frequency of static cytonemes. As a result, we obtained 'mixed' violin plots representing the distribution of the lifetime of the whole cytoneme population.

## Western blot analysis

The expression levels of the proteins induced by the UAS constructs were analyzed by western blotting (*Figure 2—figure supplement 1*). Protein extracts from third-instar larvae of *tubGal4/tubGal80$^{ts}$;UAS.ihog.RFP/+*, *tubGal4/tubGal80$^{ts}$;UAS.ihogCT-RFP/+*, *tubGal4/tubGal80$^{ts}$;UAS.ihogΔIg-RFP/+*,*tubGal4/tubGal80$^{ts}$;UAS.ihogΔFn-RFP/+*, *tubGal4/tubGal80$^{ts}$;UAS.ihogΔFn1-RFP/+*, and *tubGal4/tubGal80$^{ts}$;UAS.ihogΔFn2-RFP/+* genotypes were prepared in lysis buffer containing protease inhibitors. The samples were re-suspended in sample buffer with β-mercaptoethanol and subjected to 1 D SDS–PAGE (8%) and western blotting. Blotted membranes were probed with the anti-RFP (1:5000 rabbit polyclonal Chromotek). Blots were incubated with fluorescent α-rabbit (680RD) secondary antibody (Li-Cor) and imaged with the 364-Odyssey equipment.

## Acknowledgements

We are grateful to Ana-Citlali Gradilla and Nicole Gorfinkiel for their invaluable advice and comments on the manuscript and to Marcus Bischoff for his expertise and advice in setting up the in vivo imaging of the abdominal histoblasts in our lab and for hosting IS-B to learn to do it. We thank T Kornberg, X Zheng, P Beachy, M Bischoff, JP Vincent, and R Holmgren for stocks and reagents and to X Zheng for her help in the characterization of Ihog mutant constructs. We also thanks Transgenesis and Confocal Facilities of the CBMSO and Bloomington and Vienna stock centers for fly stocks. This work was supported by grants BFU2014-59438-P and BFU2017-83789-P and TENTACLES consortium RED2018-102411-T to IG from the Spanish Ministry of Science, Innovation and Universities and by institutional grants from the Fundación Areces and from Banco de Santander to the CBMSO. FPI fellowship from the Spanish Ministry of Science, Innovation and Universities supported CJ (BFU2017-83789-P), AA-T (BFU2014-59438-P), and IS-B (BFU2011-25987). 'Fellowships for Excellence' from the International PhD Program in Molecular Life Sciences of the Biozentrum, University of Basel supported GA.

## Additional information

### Funding

| Funder | Grant reference number | Author |
|---|---|---|
| Ministerio de Economía y Competitividad | BFU2014-59438-P | Eléanor Simon<br>Irene Seijo-Barandiarán<br>Gustavo Aguilar<br>David Sánchez-Hernández<br>Adrián Aguirre-Tamaral<br>Laura González-Méndez<br>Isabel Guerrero |
| Ministerio de Ciencia, Innovación y Universidades | BFU2017-83789-P | Eléanor Simon<br>Carlos Jiménez-Jiménez<br>David Sánchez-Hernández<br>Laura González-Méndez<br>Isabel Guerrero |
| Ministerio de Ciencia, Innovación y Universidades | RED2018-102411-T | Isabel Guerrero |

The funders had no role in study design, data collection and interpretation, or the decision to submit the work for publication.

### Author contributions

Eléanor Simon, Experimental Data, Data interpretation, Methodology, in vivo imaging, Writing and editing; Carlos Jiménez-Jiménez, Experimental data, Data interpretation, Formal analysis and quantification, in vivo imaging, Validation, Methodology, Writing - review and editing; Irene Seijo-

Barandiarán, Experimental data, Data Interpretation, Methodology, in vivo imaging, Writing - review and editing; Gustavo Aguilar, Experimental data, Tool generation, Data interpretation, Writing - review and editing; David Sánchez-Hernández, Experimental data and tool generation; Adrián Aguirre-Tamaral, Formal analysis and quantifications; Laura González-Méndez, Experimental data; Pedro Ripoll, Genetic tools, Writing - review and editing; Isabel Guerrero, Study design, Methodology, Formal analysis, Resources, Supervision, Funding acquisition, Validation, Writing - original draft, Project administration, Writing - review and editing

**Author ORCIDs**
Carlos Jiménez-Jiménez ![ORCID] https://orcid.org/0000-0001-9848-184X
Isabel Guerrero ![ORCID] https://orcid.org/0000-0001-6761-1218

**Decision letter and Author response**
Decision letter https://doi.org/10.7554/eLife.64581.sa1
Author response https://doi.org/10.7554/eLife.64581.sa2

## Additional files

**Supplementary files**
• Source code 1. R-Studio code of the statistical analysis of the Dally, Dlp, and Hh recruitment under the expression of different Ihog mutant constructs. The normality of the data was tested employing Shapiro test, and the differences between means was done using the T-test for normal data and the Wilcox test for non-normal data.

• Source code 2. Matlab script was designed to organize the data, to compute the statistical analysis, and to represent the results in different violin plots. For genotypes showing only dynamic cytonemes, the statistical analysis was done using their numerical lifetimes. To examine the normality of the data distribution, we performed a Shapiro–Wilk test. Since the results showed a non-parametric distribution of the experimental data, we selected a Wilcoxon rank sum test to compare the numerical lifetimes between two genotypes. For genotypes showing both static and non-static cytonemes we defined a no numerical case to quantify the frequency of static cytonemes. As a result, we obtained 'mixed' violin plots representing the distribution of the lifetime of the whole cytoneme population.

• Transparent reporting form

**Data availability**
All data generated or analysed during this study are included in the manuscript and supporting files.

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
