## [Decision Letter]

**Acceptance summary:**

In general, the revised version of the manuscript has improved substantially. The structure is clear, and the experiments are ordered in a logical way. The re-organisation of the results has improved readability. The authors have introduced new figures and eliminated those which were unnecessary. Overall, I think this is an excellent manuscript.

**Decision letter after peer review:**

Thank you for submitting your article "Glypicans specifically regulate Hedgehog signaling through their interaction with Ihog in cytonemes" for consideration by *eLife*. Your article has been reviewed by 3 peer reviewers, one of whom is a member of our Board of Reviewing Editors, and the evaluation has been overseen by Didier Stainier as the Senior Editor. The reviewers have opted to remain anonymous.

The reviewers have discussed the reviews with one another and the Reviewing Editor has drafted this decision to help you prepare a revised submission.

In summary, this manuscript elucidates the function of Glypicans in Hh transport via cytonemes. The reviewers felt that the manuscript describes convincingly that the fly glypicans Dally and Dally-like are required to maintain the expression of the Hh co-receptor Ihog, which stabilises cytonemes to establish the Hh gradient in the wing imaginal disc. A molecular analysis of Ihog domains was well executed.

Although the manuscript provides an in-depth analysis, the reviewers believe that the presentation of the data is rather challenging for the readers. The authors need to clearly describe the different roles that have been attributed to the glypicans and for every experiment presented, a clear explanation of the impact of the results is needed e.g. Figure 5. In addition, the stability of Ihog and Boi by altered Glypican levels and their ability to stabilize cytonemes needs to be investigated. Finally, linking the Ihog analysis to cytoneme stability analysis needs improvement.

The detailed criticisms of the reviewers are listed below:*Reviewer #1:*

In the article "Glypicans specifically regulate Hedgehog signalling through their interaction with Ihog in cytonemes" Simon et al., elucidate the function of Glypicans in Hh transport via cytonemes. The manuscript describes convincingly that the fly glypicans Dally and Dally-like are required to maintain the expression of the Hh co-receptor Ihog. Ihog – in turns – stabilises Hh cytonemes through its interaction with Glypicans to establish the Hh gradient in the wing imaginal disc. The authors further carried out an extensive molecular analysis of Ihog and identified the relevant domains within the protein required for interactions with Glypicans, Patched, and Hh. In general, this is a very thorough, detailed analysis of Ihog function. The images and videos are excellent. However, prior publication, there are two major criticisms, which needs to be addressed, in my opinion.

Firstly, the first part of the manuscript, the molecular analysis of Ihog (Figure 1-4) seems to be detached from the second cytoneme-focussed part (Figure 5, 6). Independent evidence is needed to show support for the idea that the Ihog-Gly mediated stabilisation of cytonemes is responsible for the expansion of the signalling gradient. Are the static cytonemes involved in a flattened gradient or are the receiving cells just sensitised for Hh? Can cytonemes be (de-) stabilised w/o interfering with Hh components to untangle these observations?

The authors write "Intriguingly, the same Ihog domains that regulate cytoneme dynamics are those also involved in the recruitment of Hh ligand, glypicans and the reception complex. "

My concern is that cytoneme dynamics and Hh gradient formation could be two parallel, independent events -> one needs to show this interdependency in a clear way. I could imagine an analysis of the consequences when Ihog is overexpressed, and cytoneme formation is inhibited (by other means). Consistently, could one stabilise cytonemes in an Ihog-reduced background and analyse gradient formation?

Secondly, the authors demonstrate an effect of Ihog alterations on the formation of the gradient. However, what is the physiological relevance? What are the consequences of Ihog/Gly-mediated cytoneme stabilisation and gradient formation on tissue patterning and wing formation? If this is not possible to show experimentally, this needs to be discussed.

Specific comments:

First, the authors describe the interaction between Ihog and Dally/Dally-like. They make use of the double mutants to show that Dally/Dlp are required for stabilisation of Ihog in the wing imaginal disc. Here it would be essential to strengthen the statement that Ihog is stabilised by glypicans. Can the authors provide supporting evidence e.g. biochemical ones?

Next, in an exquisite set of experiments, the authors demonstrate that the FN1 and FN2 domains are required for glypican stabilisation. The Western blot analysis – to demonstrate the different weight of the constructs- needs improvement or at least the authors need to provide alternative evidence.

The authors then analyse the interaction of Ihog with Hh and glypicans by using specific constructs with point-mutation in the FN1 or FN2 domain. This is an excellent set of data and shows very convincingly that FN1 (D558N, N559S, E561Q) is vital for Hh retention but not for interaction with glypicans. These experiments show that specific amino-acids of FN1 interact with glypicans, and these are different from those that carry out the Ihog -Hh interaction.

Next, the authors investigate the implication in Hh gradient formation. Again, the set of experiments is well documented. However, in Figure 4F, the authors show the activation of Ptc directly at the border zone. To my understanding, the interaction with Hh should not be permitted in this experiment and target genes should not be induced. So, it is unclear to me why the authors observe restricted but quite strong Ptc expression. The authors need to explain this observation. Would an independent TC approach help to elucidate signalling capability of Ihog-FN1***?

The authors claim that the stabilisation of Ihog-bearing cytonemes is glypican-dependent. The movies and the still pictures present in Figure 5 are challenging to interpret. The clone sizes are variable, and the authors claim that larger and smaller clones show different results. Furthermore, it is unclear to me why the authors use for this experiment the ttv/botv mutant instead of the dally/dlp mutant. The author should add these experiments or explain why this is not possible.

The videos accompanying Figure 6 are compelling and show the role of Ihog domains in the regulation of cytoneme dynamics, however, require a better explanation.

The discussion is well structured. However, there is data showing that glypicans can regulate secretion and stability of Hh. The authors need to discuss these findings*Reviewer #2:*

This manuscript interrogates function of Ihog and Boi adhesion molecules in cytoneme-based transport of the Hedgehog morphogen in *Drosophila*. The cell biology of how cytonemes are regulated to deliver morphogen signals is not yet well understood, so the work addresses an important topic that will be of interest to a broad audience. However, much of the study refines previous work from the same group to provide only a modest advance in understanding of how Ihog impacts cytoneme behavior.

The authors use genetic strategies in *Drosophila* to investigate how Ihog and Boi influence cytoneme dynamics. They find that the two proteins act differently with regard to cytoneme function. Boi effects are not exhaustively analyzed, but a number of genetic experiments are performed to interrogate Ihog. The authors reveal that the extracellular domains of Ihog interact with the glypicans Dally and Dlp to stabilize cytonemes that originate from Ihog over-expressing cells. Knockdown of Ihog does not alter cytoneme dynamics.

The most novel aspect of the study – that Boi functions differently than Ihog in cytonemes – is, unfortunately, not expanded upon. Some experiments lack controls or are presented in a manner that prevents clear interpretation of results. Thus, the study appears too preliminary for publication at this time.

Key points to be addressed:

Figure 1: Null alleles and RNAi silencing are used interchangeably to reduce Ihog, Boi, Dally and Dlp function in vivo. Results between methods are directly compared. Oftentimes, controls are not included to confirm the level of knockdown following RNAi. If possible use null alleles due to consistency. However, if this is not possible due to experimental reasons, give an explanation and state impact in the discussion.

Ihog levels decrease following loss of Dally or Dlp and Boi levels appear to increase following knockdown of Ihog, Dally, or Dlp. These stability changes have previously been reported. The mechanism is not clear, so should have been investigated here – especially the increased Boi protein level. How does this occur? Is stabilization occurring at the protein level or is gene expression changing? Is this a compensatory upregulation?

Based upon the supplement for Figure 2, it looks like the Ihog truncation mutants show variable stability. Might this be affecting the extent to which they alter Dally or Dlp stability? The western blot data are presented as crops of single bands adjacent to crops of a molecular weight ladder. Blots should be shown as intact images, preferable with all variants compared across a single gel with a loading control. As presented, relative stability/expression levels are impossible to assess.

Figures 3-4: Ihog mutant transgenes are tagged with either HA or RFP. Best to be consistent with tags when mutant function is being directly compared. Given that the HA tag is a small epitope and the RFP is a protein tag, they may differentially alter protein functionality. To be consistent it would be preferable to use the same tags. However, if this is not possible due to experimental reasons, the technical implication can also be mentioned in the discussion.

Figure 5: Investigation of histoblast cytonemes reaching into ttv, botv mutant clones: The ability of cytonemes to invade double mutant clones is altered only under the engineered situation of glypican dysfunction combined with Ihog over-expression. From this, it is concluded that Ihog is acting with glypicans to stabilize cytonemes. This may be the case, but they ability to see it only under an engineered situation of compound mutation plus Ihog over-expression leads this review to question the physiological relevance of the observation. Of similar concern is that the authors state the ability of Ihog over-expressing cell cytonemes to cross small vs. large ttv, botv clones differs. The difference is very difficult to appreciate from the results presented.

Figure 6: The apparent functional difference between Ihog and Boi in the ability to stabilize cytonemes is potentially very interesting, but is not investigated, which limits the advance of the current study.*Reviewer #3:*

The glypicans Dally and Dlp have important roles in morphogen signaling, and this work is of particular interest for me because it significantly advances our understanding of the multiple roles they appear to have in signal processing, signal presentation and signal reception. It is unfortunate that most of the literature has presented results and phenotypes in simplistic or simple-minded ways that do not recognize the different roles or the glypicans, or do not take experimental approaches that might distinguish them. This work of the Guerrero lab is an exception, as it is an important contribution to understanding these different roles, especially given the additional complexity introduced by the role of cytonemes. If its thoroughness and in-depth analysis are typical of work from this lab, so is the challenging presentation that makes understanding it so difficult. My recommendation to the authors is to clearly describe the different roles that have been attributed to the glypicans and for every experiment they present, clearly articulate how the results might implicate or distinguish any or several of them.

Although the figures are excellent, the manuscript is not well-written and would benefit from a complete rewrite, perhaps with the help of a native English-speaker. The following suggested changes should be taken only as examples that demand improvement.

Ln 55 "and it have been described a spatial and temporal correlations between Hh gradient" not grammatically correct.

Ln 91 "Ihog and Boi are also independently needed for the maintenance of normal Hh levels in Hh-producing cells (Yan et al., 2010)." I do not understand the meaning of "independently needed".

Ln 115 "Previous works have already described a genetic interaction between Ihog, Boi and Dlp (Yao et al., 2006)" It would help if the "genetic interactions" were explained. The Yao et al. reference (a Lis Heat Shock study) may be incorrect?

Ln 119 "Conversely, in double ttv-/- botv-/- mutant clones, responsible of synthesizing the Heparan Sulfate chains of the glypicans Dally and Dlp," This sentence reads that the mutant clones are responsible for synthesizing HS?

Ln 124 "this effect on Ihog requires of both Dally and Dlp proteins" Isn't the effect a consequence of lack of Dally and Dlp?

Ln 149 It might help if the term "build up effect" were defined and if were clearly stated that overexpression of Dally/Dlp increases Ihog levels and overexpression of Ihog increases Dally/Dlp levels.

Ln 152 The term "accretion" has a specific meaning relating to time dependence of increase that is not relevant here. Always better to use the simplest possible terms – nothing wrong with "increase"?

ln 168 Previous work from the Guerrero lab showed that extracellular Hh at the apical and basolateral plasma membranes represent different pools with respect to Hh trafficking in the Hh-producing cells. Please clarify which pool (or pools) the Ihog-driven increases represent.

Ln 177" This result, together with the fact that deletion of neither Fn1 nor Fn2 accumulates Dlp and slightly increase Dally levels (Figure 3E,E' and F,F'), suggests the Ihog Fn1-Hh recruitment also needs Ihog-glypican interaction." I do not understand this sentence.

Ln 190 "indicating that the Fn1 amino-acids that interact with glypicans are specific and different from those that carry out the Ihog-Hh interaction" I don't follow why this result shows that the interacting glypican amino acids are specific – not the ones implicated in Hh binding, so clearly different, but why specific?

Ln 213 Figure 3H?

Ln 226 "indicating that might be the presentation of Ptc to the membrane is disturbed" Is this justified given the multiple roles of Ihog/HSPGs in both Hh-producing and Hh-receiving cells?

Ln 235 "normal reception in the first cell row adjacent to the compartment border" Might this be a consequence of the low expression level of ptcGal4 in the first row?

Ln 252 "Ihog-glypicans interactions could be key regulators of cytoneme dynamics and their signaling function" Regulation is not synonymous with requirement and it is my understanding that a the glypicans have been shown to be required.

Ln 260 I am not sure what can be done to help the reader focus on the features of interest in the videos but I found the videos to be difficult if not impossible to evaluate.

Ln 275 "while glypicans might not directly influence cytoneme formation, they have an effect on their dynamics in trans, maybe regulating Ihog presence in opposing cytonemes" It is not apparent to me why the effects on cytoneme presence justifies the proposal that glypicans regulate Ihog in trans.

Ln 304 Isn't it important to distinguish between Ihog at normal levels and the phenotypes associated with Ihog overexpression?

Ln 32 "in the wing imaginal disc long cytonemes are only observed in those cases in which cytonemes are stabilized in histoblasts" Do the authors mean "conditions" (not cases)?

Ln 319 "This unexpected cytoneme behavior indicates that the absence of the Ig domain of Ihog might be affecting an interaction with an unknown extracellular matrix component also necessary for cytoneme stabilization" The authors may want to consider that the observed differences may be a consequence of quantitative rather than qualitative differences.

Ln 325 "ectopic expression of Boi fails to stabilize cytonemes" Better: over-expression of Boi does not stabilize cytonemes?

Ln 332 "Ihog induced cytoneme extension and stability is regulated by Ihog-glypican interactions" Better: Ihog induced cytoneme extension and stability is dependent on Ihog-glypican interactions?

Ln 341 "These data indicate that wild-type cells do not require these adhesion molecules to produce dynamic cytonemes." Because the data and phenotypes are based on experimental conditions that what might be revealing quantitative parameters, I do not agree that this strong conclusion is justified.

Figure 7 I do not understand why this model focuses only on the receiving cell when so much of the presented data is for the producing cells.

[Editors' note: further revisions were suggested prior to acceptance, as described below.]

Thank you for submitting your article "Glypicans define unique roles for the Hedgehog co-receptors Boi and Ihog in cytoneme-mediated gradient formation" for consideration by *eLife*. Your article has been reviewed by 3 peer reviewers, one of whom is a member of our Board of Reviewing Editors and the evaluation has been overseen by Didier Stainier as the Senior Editor. The following individual involved in review of your submission has agreed to reveal their identity: Stacey K Ogden (Reviewer #2).

The reviewers have discussed their reviews with one another, and the Reviewing Editor has drafted this to help you prepare a revised submission. All reviewers found that the manuscript was significantly improved, but also made some additional suggestions which we would like you to consider.*Reviewer #1:*

In general, the revised version of the manuscript has improved substantially. The structure is clear, and the experiments are ordered in a logical way. The re-organisation of the results has improved readability. The authors have introduced new figures and eliminated those which were unnecessary. Overall, I think this is an excellent manuscript, with only a few further suggestions.

For me, the main question remains: is iHog (specifically via the FNIII domains) directly involved in cytoneme stabilisation or indirectly via stabilisation of Hh and/or Glypicans. I feel this is an important question to discuss as in the iHog-/- situation, Glypican level on the basolateral side are reduced, which is also the side at which the cytonemes form. In contrast, in Boc mutants, glypicans still accumulate at the basal side, and therefore cytoneme formation is unaltered.

It is very much appreciated that most RNAi experiments have been removed or exchanged by induced null clones. This gives the experimental setting a compelling flow. The improved Western blot analysis is convincing (Figure 2 suppl. 1).

The authors have now tagged all constructs with RFP. This is an enormous effort and very much appreciated. The interpretation that the Fn2-RFP domain is more required explicitly for Dally interaction is good – while Fn1 is equally needed for both Dally and Dlp interaction.*Reviewer #2:*

The revised manuscript is significantly improved over the initial submission. The authors have addressed reviewer comments with new experiments, and the conclusions of the study are now well supported. I think the more in depth investigation of the differences between ihog and boi have significantly improved the study, and I support its publication in *eLife*.

*Reviewer #3:*

This revised manuscript is much improved and contributes a clear and important advance.

---

## [Author Response]

Reviewer #1:Specific comments:First, the authors describe the interaction between Ihog and Dally/Dally-like. They make use of the double mutants to show that Dally/Dlp are required for stabilisation of Ihog in the wing imaginal disc. Here it would be essential to strengthen the statement that Ihog is stabilised by glypicans. Can the authors provide supporting evidence e.g. biochemical ones?

We have tried to perform π assays between Ihog and glypicans, but we did not succeed to get a biochemical interaction. Although other groups identified a structural region located on the surface of Dlp important to mediate its role in Hh signaling, apparently the core of the purified Dlp protein did not interact appreciably with Hh or with a Hh: Ihog complex (Kim et al., 2011).

In addition, based on our results about the Ihog stabilizing effects observed in double mutant clones for *ttv*/*btv* and *dally*/*dlp* (Figure 1), we think possible that for the interaction between Ihog and glypicans to be mediated through the sugar chains of HS. These type of interactions may not be very stable and, therefore, difficult to obtain positive results through π assays.

Next, in an exquisite set of experiments, the authors demonstrate that the FN1 and FN2 domains are required for glypican stabilisation. The Western blot analysis – to demonstrate the different weight of the constructs- needs improvement or at least the authors need to provide alternative evidence.

We have done new Western blots showing clearly the different molecular weights of the Ihog mutant constructs. In the revised version of the manuscript a better Western blot is now shown in Figure 2—figure supplement 1.

Next, the authors investigate the implication in Hh gradient formation. Again, the set of experiments is well documented. However, in Figure 4F, the authors show the activation of Ptc directly at the border zone. To my understanding, the interaction with Hh should not be permitted in this experiment and target genes should not be induced. So, it is unclear to me why the authors observe restricted but quite strong Ptc expression. The authors need to explain this observation. Would an independent TC approach help to elucidate signalling capability of Ihog-FN1***?

We thank the Referee for his/her comments on this point. We have repeated the experiments several times to make sure of the activation of Ptc in the first cell row that we had presented in the original version of the manuscript. From the data obtained we conclude that this construct is acting as dominant negative, blocking all Hh responses (Ptc, En and Ci). We have also observed that after Ihog-ΔFN1and Ihog-FN1*** expression the endogenous Ihog protein is much decreased. Although there is still a very mild activation of Ptc in the first cell row, we have not given much importance to this small activation that could be caused by the possible presence of Boi.

The authors claim that the stabilisation of Ihog-bearing cytonemes is glypican-dependent. The movies and the still pictures present in Figure 5 are challenging to interpret. The clone sizes are variable, and the authors claim that larger and smaller clones show different results. Furthermore, it is unclear to me why the authors use for this experiment the ttv/botv mutant instead of the dally/dlp mutant. The author should add these experiments or explain why this is not possible.

For this work, we have considered unnecessary to induce dally/dlp double mutant clones in the histoblast nests, since it is an experiment difficult to do by life imaging. Additionally, we had previously demonstrated in wing imaginal discs the requirement of the HSPG (Bischoff et al., 2013) and of Dally and Dlp (González-Mendez et al., 2017) for the ectopic Ihog stabilizing effect on cytonemes. Therefore, we have eliminated the trans interaction of HSPG and Ihog for cytoneme stability shown originally in Figure 5.

The videos accompanying Figure 6 are compelling and show the role of Ihog domains in the regulation of cytoneme dynamics, however, require a better explanation.

We have tried to better label the presentation of the videos. We have included the appropriated labeling in the movies and we have also included the measurements of cytoneme dynamics in all mutant conditions in a new Figure 5 figure supplement 2.

The discussion is well structured. However, there is data showing that glypicans can regulate secretion and stability of Hh. The authors need to discuss these findings

In the new version of the manuscript we have discussed the functions of Dally and Dlp in Hh secretion and stability. As far as we know, the specific role of Dally in the stability of Hh in the A compartment was first proposed by Han et al., (2004) and in the P compartment by Bilioni et al., (2013). The specific role of Dlp in Hh secretion interacting with Dispatched was proposed by Callejo et al., (2011).

Reviewer #2:The authors use genetic strategies in *Drosophila* to investigate how Ihog and Boi influence cytoneme dynamics. They find that the two proteins act differently with regard to cytoneme function. Boi effects are not exhaustively analyzed, but a number of genetic experiments are performed to interrogate Ihog. The authors reveal that the extracellular domains of Ihog interact with the glypicans Dally and Dlp to stabilize cytonemes that originate from Ihog over-expressing cells. Knockdown of Ihog does not alter cytoneme dynamics.The most novel aspect of the study – that Boi functions differently than Ihog in cytonemes – is, unfortunately, not expanded upon.

We thank the Referee for this comment that has allowed us to further explore these differential functions of Boi and Ihog in Hh signaling. Since we agreed with the referee that this was the most novel aspect of our work, we reoriented the writing of the manuscript basing it on the new results about the differential roles of Boi and Ihog on Hh signaling. Accordingly, we have also changed the title of the manuscript in this regard.

Some experiments lack controls or are presented in a manner that prevents clear interpretation of results. Thus, the study appears too preliminary for publication at this time.

We agree with the Referee that the presentation of some of the results appeared too preliminary. We have tried to improve the presentation of our data by expanding further our results and introducing the appropriated controls and quantifications. We hope that this revised version of the manuscript has been improved in all these aspects.

Key points to be addressed:Figure 1: Null alleles and RNAi silencing are used interchangeably to reduce Ihog, Boi, Dally and Dlp function in vivo. Results between methods are directly compared. Oftentimes, controls are not included to confirm the level of knockdown following RNAi. If possible use null alleles due to consistency. However, if this is not possible due to experimental reasons, give an explanation and state impact in the discussion.

Since the Referee’s concern is appropriate, we have eliminated the knockdowns following RNAi expression in Figure 1. We have now induced null clones and analyzed the expression of Ihog and Boi in all cases. Therefore, the old Figure 1 figure supplementary 1 has been eliminated in the revised version of the manuscript.

However, the expression of RNAis to silence Ihog and Boi were used in the new Figures 5 and 6. The controls to confirm the knockdown of Ihog and Boi following expression of the RNAis are now shown in Figure 5 figure supplementary 1.

Ihog levels decrease following loss of Dally or Dlp and Boi levels appear to increase following knockdown of Ihog, Dally, or Dlp. These stability changes have previously been reported. The mechanism is not clear, so should have been investigated here – especially the increased Boi protein level. How does this occur? Is stabilization occurring at the protein level or is gene expression changing? Is this a compensatory upregulation?

In fact, we had previously investigated the cause of the slight increase of Boi levels after knocking down Ihog. We had found that it was at the protein level because no *boi* transcriptional changes were observed using a transcriptional reporter. This is possibly due to compensatory upregulation, but we have not investigated this further.

Based upon the supplement for Figure 2, it looks like the Ihog truncation mutants show variable stability. Might this be affecting the extent to which they alter Dally or Dlp stability?

All Ihog mutant forms show similar levels of expression with the exception of IhogΔIg, which shows lower levels but still accumulate Hh, Dlp and Dally as the full length Ihog. However, the cytonemes are as dynamic as the expression of Ihog FN2 mutants. Therefore, we still think that the differential effects on Dally and Dlp after expressing all mutant forms are specific.

The western blot data are presented as crops of single bands adjacent to crops of a molecular weight ladder. Blots should be shown as intact images, preferable with all variants compared across a single gel with a loading control. As presented, relative stability/expression levels are impossible to assess.

We have done new Western blots, which are presented in the new Figure 2 figure supplement 1.

Figures 3-4: Ihog mutant transgenes are tagged with either HA or RFP. Best to be consistent with tags when mutant function is being directly compared. Given that the HA tag is a small epitope and the RFP is a protein tag, they may differentially alter protein functionality. To be consistent it would be preferable to use the same tags. However, if this is not possible due to experimental reasons, the technical implication can also be mentioned in the discussion.

Following the Referee’s suggestion and to be consistent with the use of the same tag in all the constructs, we made another ΔFn2 transgene tagged to RFP. Besides, Dfn2HA and DFn2-RFP behave similarly regarding the effects concerning Hh interaction, gradient formation and cytoneme dynamics. However, following the quantification of the effects on the accumulation of Dally and Dlp we found that the Fn2-RFP domain is more specifically required for Dally interaction while Fn1 is equally needed for both Dally and Dlp interaction. Only the results obtained using the DFn2-RFP and not of DFn2-HA are presented in the revised version of the manuscript.

Figure 5: Investigation of histoblast cytonemes reaching into ttv, botv mutant clones: The ability of cytonemes to invade double mutant clones is altered only under the engineered situation of glypican dysfunction combined with Ihog over-expression. From this, it is concluded that Ihog is acting with glypicans to stabilize cytonemes. This may be the case, but they ability to see it only under an engineered situation of compound mutation plus Ihog over-expression leads this review to question the physiological relevance of the observation. Of similar concern is that the authors state the ability of Ihog over-expressing cell cytonemes to cross small vs. large ttv, botv clones differs. The difference is very difficult to appreciate from the results presented.

We agree with the referee that the presentation of the results in the old Figure 5 was a bit confusing. Since we had previously demonstrated in wing imaginal discs the requirement of the HSPG (Bischoff et al., 2013) and of the Dally and Dlp (González-Mendez et al., 2017) for the Ihog stabilizing effect on cytonemes, we have eliminated the interaction of HSPG and Ihog shown in the old Figure 5. Obviously, it would have been better to induce *dally/dlp* double mutant clones in the histoblast nests, but this is an experiment difficult to do by life imaging.

Figure 6: The apparent functional difference between Ihog and Boi in the ability to stabilize cytonemes is potentially very interesting, but is not investigated, which limits the advance of the current study.

We thank the Referee for his/her suggesting and agree that the apparent functional difference between Ihog and Boi was potentially very interesting. We have explored further the interaction of Boi with the glypicans and obtained interesting results that made us to describe the different requirements of Boi and Ihog for the formation of the Hh gradient. The revised version of the manuscript is centered on these results.

Reviewer #3:Although the figures are excellent, the manuscript is not well-written and would benefit from a complete rewrite, perhaps with the help of a native English-speaker.

We thank the Referee for his/her comments and for the enormous effort in correcting our manuscript. We agree that the original manuscript was not clearly written, the results were not very well organized, and some figures were presented a bit preliminary. We hope that the revised version of the manuscript is improved. We have reorganized our results, introduced new figures and eliminated what was considered unnecessary.

Ln 55 "and it have been described a spatial and temporal correlations between Hh gradient" not grammatically correct

It has been corrected

Ln 91 "Ihog and Boi are also independently needed for the maintenance of normal Hh levels in Hh-producing cells (Yan et al., 2010)." I do not understand the meaning of "independently needed"

It has been corrected

Ln 115 "Previous works have already described a genetic interaction between Ihog, Boi and Dlp (Yao et al., 2006)" It would help if the "genetic interactions" were explained. The Yao et al. reference (a Lis Heat Shock study) may be incorrect?

We have changed the sentence and corrected the error in the reference.

Ln 119 "Conversely, in double ttv-/- botv-/- mutant clones, responsible of synthesizing the Heparan Sulfate chains of the glypicans Dally and Dlp," This sentence reads that the mutant clones are responsible for synthesizing HS?

The sentence has been corrected.

Ln 124 "this effect on Ihog requires of both Dally and Dlp proteins" Isn't the effect a consequence of lack of Dally and Dlp?

The mistake has been corrected.

Ln 149 It might help if the term "build up effect" were defined and if were clearly stated that overexpression of Dally/Dlp increases Ihog levels and overexpression of Ihog increases Dally/Dlp levels

We agree and the term “build up” has been eliminated in the revised version.

Ln 152 The term "accretion" has a specific meaning relating to time dependence of increase that is not relevant here. Always better to use the simplest possible terms – nothing wrong with "increase"?

Accretion has been changed for increase or accumulation in all places.

ln 168 Previous work from the Guerrero lab showed that extracellular Hh at the apical and basolateral plasma membranes represent different pools with respect to Hh trafficking in the Hh-producing cells. Please clarify which pool (or pools) the Ihog-driven increases represent.

We thank the referee for enquiring about the apica/lbasal distribution of the Hh pathway components. The analysis of the differential apical o basal accumulation of Dally and Dlp in the disc epithelium after the ectopic expression of Ihog and Boi indicates, together with the rest of the data, a differential requirement of these two proteins: Ihog more basal and in cytonemes while Boi more apical and not able to regulate cytoneme dynamics. These experiments indicate a unique function of each of the two Hh co-receptors, being Ihog the one needed for cytoneme-mediated long-range Hh gradient formation.

Ln 177" This result, together with the fact that deletion of neither Fn1 nor Fn2 accumulates Dlp and slightly increase Dally levels (Figure 3E,E' and F,F'), suggests the Ihog Fn1-Hh recruitment also needs Ihog-glypican interaction." I do not understand this sentence.

We have changed the sentence, since the results about the role of Fn1 and Fn2 in the glypican interactions are clearer in the new version of the manuscript.

Ln 190 "indicating that the Fn1 amino-acids that interact with glypicans are specific and different from those that carry out the Ihog-Hh interaction" I don't follow why this result shows that the interacting glypican amino acids are specific – not the ones implicated in Hh binding, so clearly different, but why specific?

We have changed this confusing sentence.

Ln 213 Figure 3H?

The numbers that correspond to each figure have been changed.

Ln 226 "indicating that might be the presentation of Ptc to the membrane is disturbed" Is this justified given the multiple roles of Ihog/HSPGs in both Hh-producing and Hh-receiving cells?

Since the result on Hh gradient was analyzed expressing the constructs in the A compartment we cannot argue about their effect in the Hh presenting cells.

Ln 235 "normal reception in the first cell row adjacent to the compartment border" Might this be a consequence of the low expression level of ptcGal4 in the first row?

We have repeated the experiments several times to make sure of the activation of Ptc in the first cell row that we had presented in the original version of the manuscript. From the data obtained we conclude that this construct is acting as dominant negative, blocking all Hh responses (Ptc, En and Ci). We have also observed that after Ihog-ΔFN1and Ihog-FN1*** expression the endogenous Ihog protein is much decreased. Although there is still a very mild activation of Ptc in the first cell row, we have not given much importance to this small activation that could be caused by the possible presence of Boi. The panels of the corresponding old version of the figure have been changed and new data have been introduced in a new Figure 4.

Ln 252 "Ihog-glypicans interactions could be key regulators of cytoneme dynamics and their signaling function" Regulation is not synonymous with requirement and it is my understanding that a the glypicans have been shown to be required.

Our experiments shown in the old Figure 5 indicated that cytonemes were still formed in the absence of glypicans but they were required for Ihog induced cytoneme stabilization. Therefore, we can only say that Ihog-glypicans interactions could be key regulators of cytoneme dynamics and their signaling function.

Ln 260 I am not sure what can be done to help the reader focus on the features of interest in the videos but I found the videos to be difficult if not impossible to evaluate.

We have tried to better label the presentation of the videos. We have included the appropriated labeling in the movies with arrows to follow cytoneme dynamics and we have also included the measurements of cytoneme dynamics in all mutant conditions in a new Figure 5 figure supplement 2.

Ln 275 "while glypicans might not directly influence cytoneme formation, they have an effect on their dynamics in trans, maybe regulating Ihog presence in opposing cytonemes" It is not apparent to me why the effects on cytoneme presence justifies the proposal that glypicans regulate Ihog in trans.

The role of glypicans as regulators of Ihog presence in cytonemes in trans has been eliminated from the results and discussion. The old Figure 5 showed that cytonemes were formed and navigated with a normal dynamics when encounter double ttv/btv mutant clones (we called this a trans interaction). Since we consider that the cis and trans interaction of Ihog and glypicans is confusing and it is not the main message of our work we decided to eliminate these data. Besides, that the HSPG were required in trans for the presence of ectopic Ihog in cytonemes in wing discs was already demonstrated in Bischoff et al., 2013.

Ln 304 Isn't it important to distinguish between Ihog at normal levels and the phenotypes associated with Ihog overexpression?

All cytonemes labelled with GMA-GFP are also marked with Ihog-RFP or with the different Ihog-RFP mutant forms in the same vidoe (González-Méndez, Seijo-Barandiarán and Guerrero, 2017). Unfortunately, it is impossible to distinguish between cytonemes expressing Ihog at normal levels and cytonemes overexpressing the Ihog or the Ihog mutant forms in the same video.

Ln 32 "in the wing imaginal disc long cytonemes are only observed in those cases in which cytonemes are stabilized in histoblasts" Do the authors mean "conditions" (not cases)?

The Referee is right, we meant conditions.

Ln 319 "This unexpected cytoneme behavior indicates that the absence of the Ig domain of Ihog might be affecting an interaction with an unknown extracellular matrix component also necessary for cytoneme stabilization" The authors may want to consider that the observed differences may be a consequence of quantitative rather than qualitative differences.

It may be possible that the effect on dynamics could be a consequence of the lower expression levels of this Ig domain deficient construct. We have introduced this possibility in the discussion of the new version of the manuscript.

Ln 325 "ectopic expression of Boi fails to stabilize cytonemes" Better: over-expression of Boi does not stabilize cytonemes?

It has been changed.

Ln 332 "Ihog induced cytoneme extension and stability is regulated by Ihog-glypican interactions" Better: Ihog induced cytoneme extension and stability is dependent on Ihog-glypican interactions?

The sentence has been corrected.

Ln 341 "These data indicate that wild-type cells do not require these adhesion molecules to produce dynamic cytonemes." Because the data and phenotypes are based on experimental conditions that what might be revealing quantitative parameters, I do not agree that this strong conclusion is justified.

Maybe this sentence is not justified. However, if this is the case, we do not think that it would be a major problem. The dynamic of cytonemes could be modulated by multiple intrinsic and extrinsic factors.

Figure 7 I do not understand why this model focuses only on the receiving cell when so much of the presented data is for the producing cells.

We do not agree with the Referee in this point. The work is focused on both receiving and producing cells.

The effect on glypicans is analyzed in both compartments, in random clones or using ApGal4. The effect on Hh in the P compartment was also analyzed using ApGal4. The effect on dynamics is presented using Hh Gal4 but the effect is similar when using Ptc-Gal4. Finally, the effect on Hh reception was done expressing the Ihog mutants in the receiving cells. Altogether these results show that ectopic Ihog induced cytoneme

stability is dependent on Ihog-glypican interactions and that the two Ihog FNIII domains (which are also responsible for the Ihog-glypicans interaction) have a predominant role in this stabilization.

[Editors' note: further revisions were suggested prior to acceptance, as described below.]

Reviewer #1:For me, the main question remains: is iHog (specifically via the FNIII domains) directly involved in cytoneme stabilisation or indirectly via stabilisation of Hh and/or Glypicans. I feel this is an important question to discuss as in the iHog-/- situation, Glypican level on the basolateral side are reduced, which is also the side at which the cytonemes form. In contrast, in Boc mutants, glypicans still accumulate at the basal side, and therefore cytoneme formation is unaltered.

We conclude that the FNIII domains of Ihog recruit glypicans at the basolateral side of the epithelium where cytonemes are formed. This Ihog interaction with glypicans could regulate cytoneme behavior and these interactions provide glypican specificity for Hh signaling as well as a key function for Hh gradient formation.